# Large-scale whole-exome sequencing analyses identified protein-coding variants associated with immune-mediated diseases in 350,770 adults

Liu Yang[1,6], Ya-Nan Ou[2,6], Bang-Sheng Wu[1,6], Wei-Shi Liu[1], Yue-Ting Deng [1], Xiao-Yu He[1], Yi-Lin Chen[1], Jujiao Kang [3,4], Chen-Jie Fei[1], Ying Zhu [5], Lan Tan[2], Qiang Dong [1], Jianfeng Feng [3,4], Wei Cheng [1,3,4] & Jin-Tai Yu [1] ✉

The genetic contribution of protein-coding variants to immune-mediated diseases (IMDs) remains underexplored. Through whole exome sequencing of 40 IMDs in 350,770 UK Biobank participants, we identified 162 unique genes in 35 IMDs, among which 124 were novel genes. Several genes, including *FLG* which is associated with atopic dermatitis and asthma, showed converging evidence from both rare and common variants. 91 genes exerted significant effects on longitudinal outcomes (interquartile range of Hazard Ratio: 1.12-5.89). Mendelian randomization identified five causal genes, of which four were approved drug targets (*CDSN*, *DDR1*, *LTA*, and *IL18BP*). Proteomic analysis indicated that mutations associated with specific IMDs might also affect protein expression in other IMDs. For example, *DXO* (celiac disease-related gene) and *PSMB9* (alopecia areata-related gene) could modulate CDSN (autoimmune hypothyroidism-, psoriasis-, asthma-, and Graves' disease-related gene) expression. Identified genes predominantly impact immune and biochemical processes, and can be clustered into pathways of immune-related, urate metabolism, and antigen processing. Our findings identified protein-coding variants which are the key to IMDs pathogenesis and provided new insights into tailored innovative therapies.

Characterized by high mortality rates[1,2] and rising prevalences[3,4], immune-mediated diseases (IMDs) pose a huge challenge to human health, yet a significant proportion lacked effective treatments[5,6]. Since they exhibit substantial heritability (around 50%)[7], unraveling the genetic architecture of IMDs might illuminate potential therapeutic strategies. However, current genome-wide association studies (GWASs) could not fully depict the entire human hereditary landscape[8], particularly rare IMDs. This limitation is further accentuated as many GWAS-identified alleles fall outside protein-coding regions, complicating identifications of direct gene-disease

[1]Department of Neurology and National Center for Neurological Disorders, Huashan Hospital, State Key Laboratory of Medical Neurobiology and MOE Frontiers Center for Brain Science, Shanghai Medical College, Fudan University, Shanghai 200040, China. [2]Department of Neurology, Qingdao Municipal Hospital, Qingdao University, Qingdao 266071, China. [3]Institute of Science and Technology for Brain-Inspired Intelligence, Fudan University, Shanghai 200443, China. [4]Key Laboratory of Computational Neuroscience and Brain-Inspired Intelligence, Fudan University, Ministry of Education, Shanghai, China. [5]State Key Laboratory of Medical Neurobiology and MOE Frontiers Center for Brain Science, Institutes of Brain Science, Fudan University, Shanghai 200032, China. [6]These authors contributed equally: Liu Yang, Ya-Nan Ou, Bang-Sheng Wu. ✉e-mail: jintai_yu@fudan.edu.cn

associations. Compared to GWAS, whole-exome sequencing (WES) focused on protein-coding regions, potentially unmasking variants directly associated with IMDs[9,10].

Identifying causal variants and their underlying biological insights can deepen our understanding of diseases and further tailor therapeutic targets. Several previous exome studies have revealed mutations in thrombotic thrombocytopenic purpura[11], psoriasis[12], and sarcoidosis (SD)[13], yet they were limited by small sample sizes, affecting their statistical robustness. It has left significant lacunae in the comprehension of protein-coding variants, not to mention the broader landscape of IMDs that remains uncharted. Moreover, previous studies have primarily focused on individual variant-disease associations rather than gene-disease relationships[14]. Despite exonic variants being protein-coding, their mutations often exert minimal functional consequences[15,16]. The strategy that involves gene-level collapsing variants, especially those predicted loss-of-function (pLOF) or predicted deleterious missense (pmis), to identify promising genetic-disease associations[17] has emerged. Third, even large-scale sequencing studies, such as that on Crohn's disease[18] or osteoporosis[19], could not fully capture the multifaceted pathophysiology and clinical implications of these variants. The pathophysiological characteristics of these diseases are crucial, as they can guide tailored therapeutic innovative therapies.

The UK Biobank (UKB), rich in multi-omic data, stands as an ideal platform for such endeavors and has been widely used for sequencing studies of human diseases and traits[7–12]. Prior investigations within the UKB, primarily through GWAS, have focused on genetic risk factors[20,21], ethnic disparities[22], identification of IMD therapeutic targets[23–25], associated comorbidities[26,27], and their unique or overlapping genetic landscape[28]. However, WES studies of IMDs in the UKB have been sparse. Some have targeted specific regions, such as the HLA region for 11 autoimmune diseases[29], or have investigated asthma risk mutations among predetermined variants[30]. Additionally, many of these studies were constrained by their sample sizes; for instance, one identified the *TET2* mutation as a risk factor for gout among only 170,000 participants[31].

In this study, by leveraging the large-scale WES data of 350,770 UKB adults, our objectives are to: (1) identify putative variants across 40 IMDs at an exome-wide level; (2) elucidate clinical impacts, and underlying biological pathways of these variants and to identify potential therapeutic targets through time-to-event analysis, Mendelian randomization (MR) analysis, proteomic analysis, and phenome-wide association analysis (PheWAS). A detailed study design is depicted in Fig. 1.

## Results

### Population characteristics

We analyzed a total of 350,770 European-descent individuals (mean age, 56.9 years; female sex, 162,210 [46.2%]) from the UKB for whom both WES data and phenotype data for IMDs were available (Supplementary Data 1). A total of 20,155,842 distinct autosomal genetic variants were available from the exome sequencing data after quality control (QC), 20,056,064 of which displayed a minor allele frequency (MAF) below 0.1%. We annotated 744,255 pLOF variants and 1,437,627 pmis variants.

### Exome-wide rare variant analysis of 30 IMDs identified 92 genes

We performed a gene-based, whole-exome-wide association analysis using the SKAT method for rare mutations (MAF < 1%) across 40 IMDs, adjusting for age, sex, the first 10 genomic principal components (PCs), and a sparse genetic relationship matrix. The Q-Q plots were presented in Supplementary Fig. 1. We incorporated four max-MAF cutoffs (maximum MAF: 1%, 0.1%, 0.001%, and 0.0001%) and two functional annotation sets (pLOF only, and pLOF+pmis) to identify new gene-phenotype associations[32]. In total, we identified 92 significant

genes across 30 IMDs at an exome-wide significance threshold of $P < 2.5 \times 10^{-6}$ (Fig. 2A and Supplementary Data 2), with 83 being unreported. Our research has discerned both previously established genes and a plethora of novel genes. Concordant with a previous research[33], we found that *FLG* (OR: 1.01, $P = 1.79 \times 10^{-6}$) was associated with atopic dermatitis (AD), and we also identified a novel gene-*STAT5B* with a more pronounced coefficient (1.16, $P = 1.18 \times 10^{-6}$). Aligned with a prior study[34], we found that *VEGFA* was associated with multiple sclerosis (MS, 1.10, $P = 6.04 \times 10^{-7}$), but our study also unraveled two novel genes, *LRRC74A* (1.08, $P = 4.20 \times 10^{-7}$) and *ZNF266* (1.08, $P = 1.10 \times 0^{-7}$). Most gene-disease pairs displayed deleterious associations, except *IL33*-asthma (0.99, $P = 8.09 \times 10^{-16}$) and *IFIH1*-psoriasis (0.98, $P = 1.02 \times 10^{-6}$).

We next sought to determine whether the signals from the detected rare variants stood independently from nearby GWAS signals. Initial analysis was carried out to identify leading signals located within ±500 kilobases (kb) from the gene. Following this, we conducted rare variant association assessments by integrating the leading signals as supplementary covariates ("Methods" section). In general, the magnitude of effects and associated P-values demonstrated minimal attenuation after accounting for GWAS signals (Supplementary Data 3). Notably, the correlation of *ABCG2* with gout ($P = 5.39 \times 10^{-8}$) and *DXO* with celiac disease ($P = 1.40 \times 10^{-7}$) displayed enhanced significance.

In order to validate the gene-based associations in UKB, we searched from Kurki et al.'s summary statistics analyzed from FinnGen dataset[35] ("Methods" section). Of the 30 disease phenotypes identified in the UKB, 24 were available in FinnGen, which covered 69 of the 92 identified genes. Searches yielded 13 associations (19% replicated) of Bonferroni-corrected significance ($P < 1.43 \times 10^{-3}$; Supplementary Data 2). Notably, *FLG* for AD (OR: 2.09, $P = 6.54 \times 10^{-53}$), *ETV7* for Primary biliary cirrhosis (PBC, 1.98, $P = 2.21 \times 10^{-4}$), and *ABCG2* for gout (1.69, $P = 2.73 \times 10^{-76}$) emerged with the most pronounced coefficients, while protective effects were also discerned for *IL33*-asthma (0.91, $P = 4.12 \times 10^{-25}$) and *IFIH1*-psoriasis (0.70, $P = 1.81 \times 10^{-8}$). The results of multi-ancestry were presented in Supplementary Data 4, with 10 association pairs in the Asian population (including *ABCG2*-gout, and *DXO*-celiac disease), and 5 in the Black population (*FLG*-AD and *IFIH1*-psoriasis) being replicated.

To offer a clinical perspective on mutations, we classified variants into pLOF and four pmis categories based on REVEL scores (75–100, 50–75, 25–50, 0–25). Using case-control enrichment as our core metric, we derived ORs using penetrance and prevalence, favoring its straightforward computation and alignment with clinical indicators over burden tests ("Methods" section). We found a significant enrichment of mutations in IMD cases within the pLOF category, exemplified by *STAT5B* and *CD28* (Fig. 2B). Relatively deleterious pmis mutations with a REVEL score of >50 were also notably enriched, such as *LRRC74A* and *AJAP1* (Fig. 2B). Collectively, 80% of the observed case enrichment was attributed to pLOF or more damaging pmis variants (Fig. 2C). The distribution pattern underscores the importance of focusing on pLOF and pmis variants with a high REVEL score (Fig. 2D). Detailed results could be found in Supplementary Data 5.

### Exome-wide common variant analysis of 20 IMDs identified 73 genes

Exome-wide common variant (MAF ≥ 1%) association analyses were conducted using PLINK v2[36]. We identified significant hits for 20 of the 40 IMDs for common mutations under the conventional threshold ($P < 5 \times 10^{-8}$). A total of 73 genes (115 gene-disease associations, Fig. 3A), among which 78 gene-disease associations were novel (Supplementary Data 6). Celiac disease was identified to have the most single nucleotide polymorphisms (SNPs, $N = 30$), followed by psoriasis ($N = 16$) and asthma ($N = 13$). The mean values for disease-predisposing and disease-

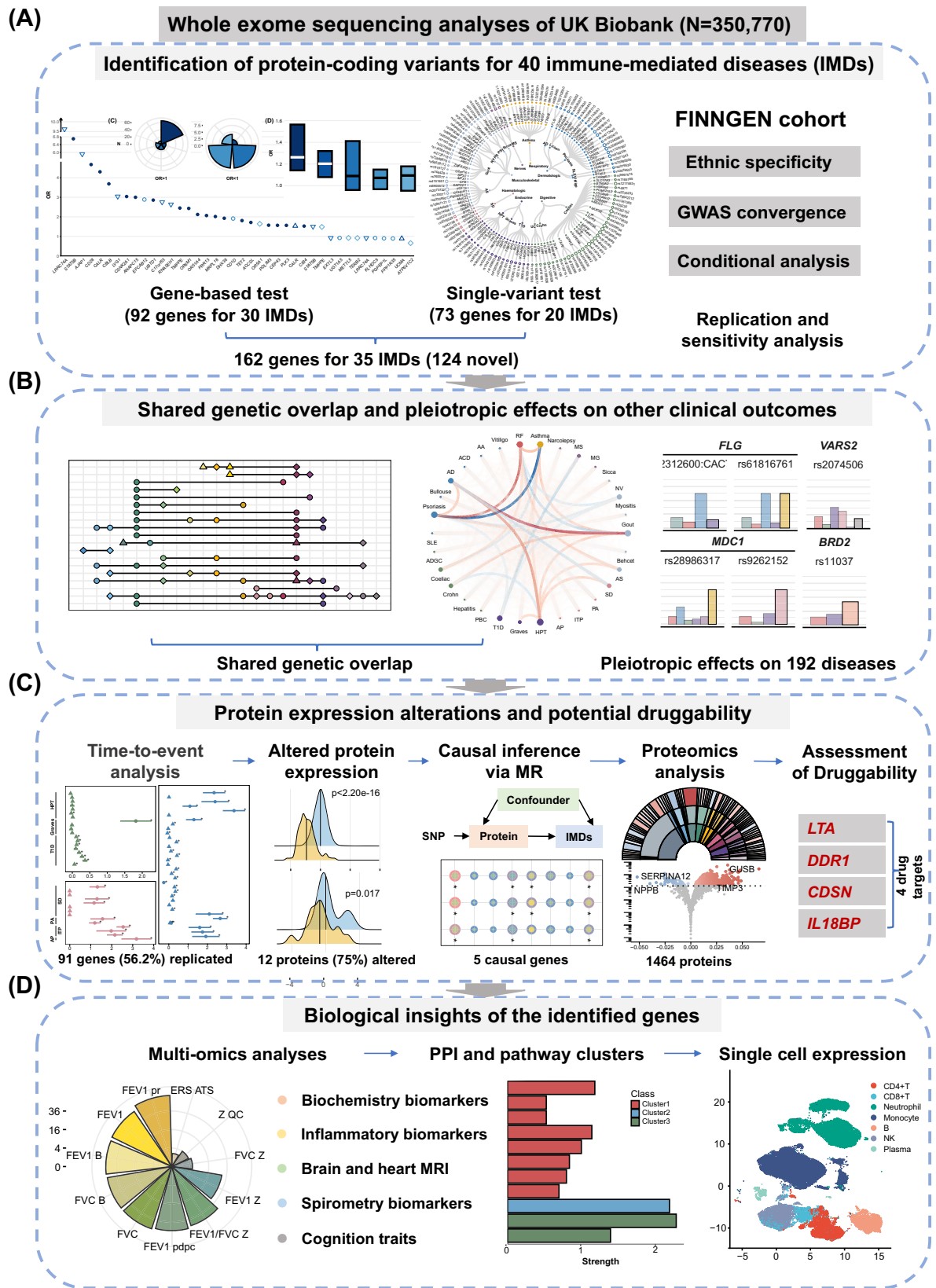

**(A)**

**Whole exome sequencing analyses of UK Biobank (N=350,770)**

**Identification of protein-coding variants for 40 immune-mediated diseases (IMDs)**

**FINNGEN cohort**

Ethnic specificity

GWAS convergence

Conditional analysis

Replication and sensitivity analysis

**Gene-based test (92 genes for 30 IMDs)**      **Single-variant test (73 genes for 20 IMDs)**

**162 genes for 35 IMDs (124 novel)**

**(B)**

**Shared genetic overlap and pleiotropic effects on other clinical outcomes**

**Shared genetic overlap**      **Pleiotropic effects on 192 diseases**

**(C)**

**Protein expression alterations and potential druggability**

**Time-to-event analysis** → **Altered protein expression** → **Causal inference via MR** → **Proteomics analysis** → **Assessment of Druggability**

Confounder

SNP → Protein → IMDs

*LTA*
*DDR1*
*CDSN*
*IL18BP*

4 drug targets

**91 genes (56.2%) replicated**   **12 proteins (75%) altered**   **5 causal genes**   **1464 proteins**

**(D)**

**Biological insights of the identified genes**

**Multi-omics analyses** → **PPI and pathway clusters** → **Single cell expression**

● Biochemistry biomarkers
● Inflammatory biomarkers
● Brain and heart MRI
● Spirometry biomarkers
● Cognition traits

protective alleles were 1.47 and 0.70, respectively. Among the risk genes, *LTA* notably escalated the risk for celiac disease (OR: 1.66, $P = 3.30 \times 10^{-43}$) and *CDSN* significantly elevated the risk of psoriasis (1.43, $P = 1.20 \times 10^{-145}$). Among the protective genes, *BTN3A2* (0.80, $P = 4.60 \times 10^{-8}$) and *DDR1* (0.79 $P = 4.37 \times 10^{-9}$) protected against SD and Graves' disease, respectively. Different from positional annotation

conducted by ANNOVAR, we also combined positional mapping with eQTL and Chromatin interaction mapping by FUMA to find the gene that they regulate. A total of 489 genes were mapped by positional or eQTL or Chromatin interaction mapping and 109 gene-disease associations were consistently mapped (94.8% overlapped with ANNOVAR; Supplementary Data 7 and 8).

**Fig. 1 | Guideline of the study.** The analytical workflow and key findings are presented in this figure. **A** Incorporating WES data of UKB, we first performed exome-wide analysis across 40 IMDs. Top left and top middle panel depicted exome-wide gene-based analysis and single-variant analysis, respectively. External replication, internal replication, and sensitive analysis were further performed (the top right panel). **B** Then, we estimated genetic overlap across IMDs (the second left panel), pairwise genetic correlations (the second middle panel), and pleiotropic effects (the second right panel) at exome level. **C** We further investigated the clinical implications of IMD-associated genes (the third panel). Panel one, quantifying longitudinal disease risks for putatively pathogenic variations; panel two and three, protein expression alteration contributed by genetic mutations and causal

inference; panel four, proteomic-wide analysis between identified genes and 1464 proteins; panel five, retrieving the potential druggability of genes by querying databases. **D** Finally, we unraveled the underlying biological insights of IMD-associated genes (the last panel). Last left panel, the associations between the IMDs-associated genes with muti-omics traits; last medium panel, shared biological pathways among identified genes; last right panel, the expression of the identified genes in different cell types. WES whole-exome sequencing, UKB the United Kingdom Biobank, IMD immune-mediated disease, GWAS genome-wide association study, MR Mendelian randomization, PPI protein-protein interaction, MRI magnetic resonance imaging.

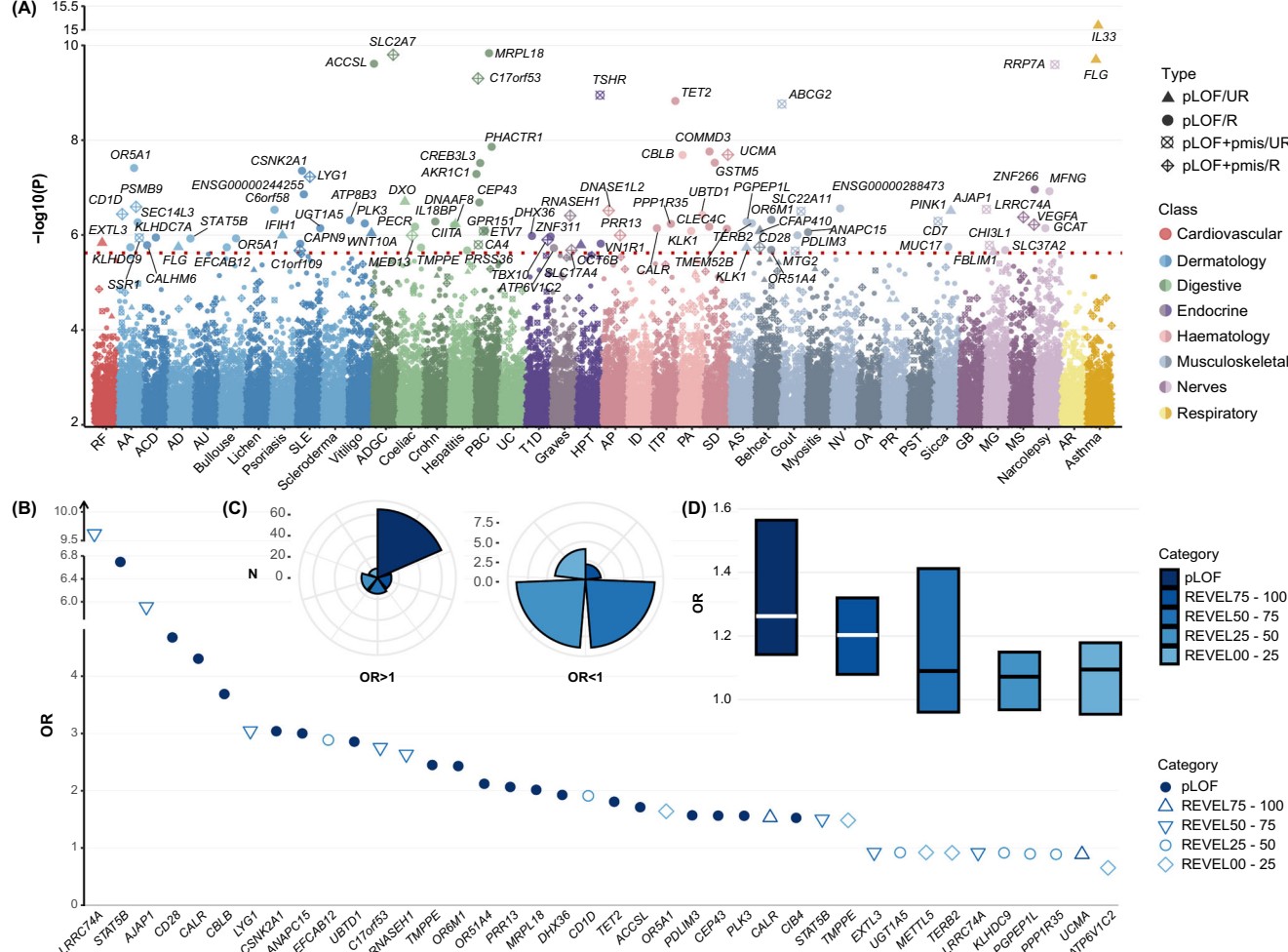

**Fig. 2 | Exome-wide analysis of rare genetic variation for 40 IMDs in UKB.**
**A** Multiple-trait Manhattan plot representing the results from exome-wide gene-based tests for each IMDs. The red dotted line indicates the significance threshold at $2.5 \times 10^{-6}$. SKAT *P*-values are two-sided and unadjusted. The significance threshold is set at FDR-corrected *P*-value < 0.05 for multiple comparisons. **B** Case-control enrichment of rare protein-coding variants in identified genes across consequence categories. The dot represents the OR; the putatively damaging nature of the variants reduces from dark blue to light blue according to the legend. **C** The number of predicted functional consequences, represented by color, that are displayed in risk-enhancing (OR > 1) and protective (OR < 1) associations. **D** Box plots of ORs in the five predicted functional consequences categories. Data are presented as median values, denoted by horizontal lines within boxes that represent the 25th and 75th percentiles, encompassing the interquartile range. Participant count: 350,770, encompassing a diverse cohort. Gene distribution is as follows: 90 genes with pLOF mutations; 45 genes with REVEL scores 75–100; 56 genes with scores 50–75; 60 genes with scores 25–50; and 50 genes with scores 0–25. IMD

immune-mediated disease, UKB the United Kingdom Biobank, pLOF predicted loss-of-function, PC principal component, FDR false discovery rate, OR odds ratio, UR ultra-rare, R rare, pmis predicted deleterious missense, REVEL rare exome variant ensemble learner, RF rheumatic fever, AA alopecia areata, ACD allergic contact dermatitis, AD atopic dermatitis, AU allergic urticaria, Bullouse bullouse disorders, Lichen Lichen planus, SLE systemic lupus erythematosus, ADGC allergic and dietetic gastro-enteritis and colitis, Celiac celiac disease, Crohn Crohn's disease, Hepatitis autoimmune hepatitis, PBC primary biliary cirrhosis, UC ulcerative colitis, T1D diabetes mellitus (Type I), Graves Graves' disease, HPT autoimmune hypothyroidism, AP allergic purpura, ID immunodeficiency with predominantly antibody defects, ITP idiopathic thrombocytopenic purpura, PA pernicious anemia, SD sarcoidosis, AS ankylosing spondylitis, Behcet Behcet's disease, NV necrotizing vasculopathies, OA osteoarthritis, PR polymyalgia rheumatica, PST psoriatic and enteropathic arthropathies, Sicca Sicca syndrome (Sjogren's syndrome); GB Guillain-Barre syndrome, MG myasthenia gravis, MS multiple sclerosis, AR allergic rhinitis.

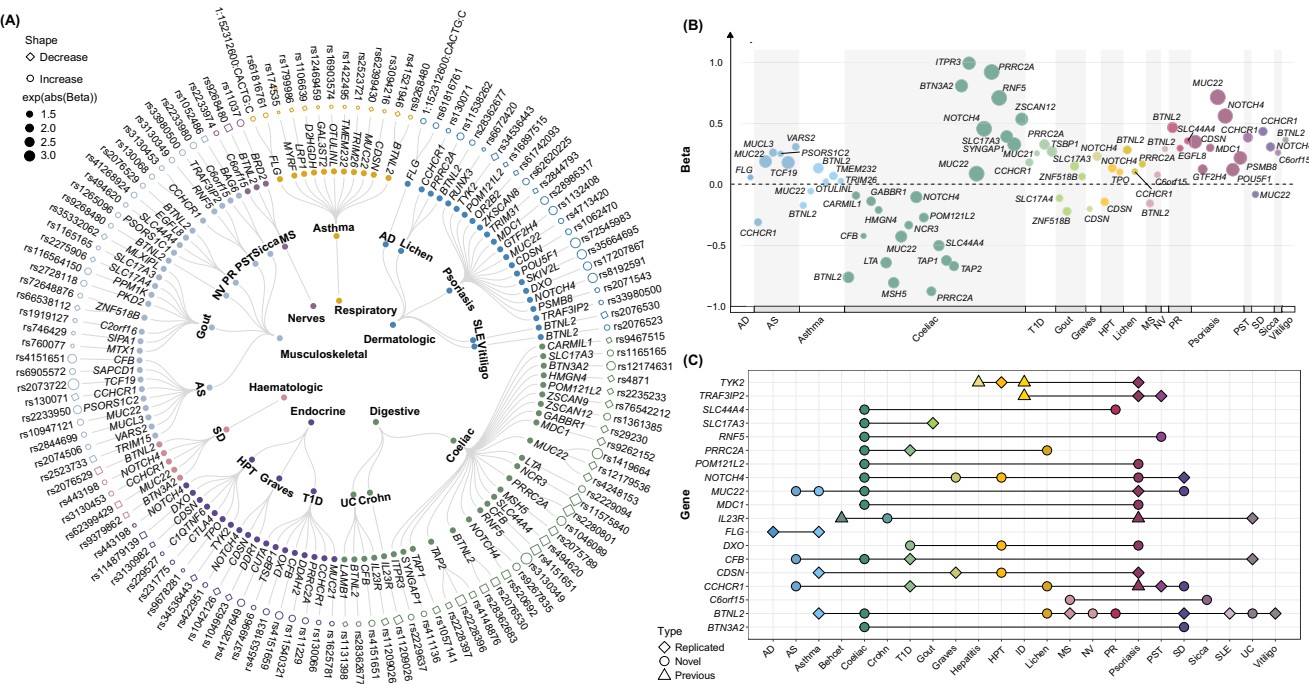

**Fig. 3 | Exome-wide analysis of common genetic variation for 40 IMDs in UKB.** **A** A circos plot representing significant associations between common genetic variants (MAF > 1%), their reference genes, and linked IMDs. The shape of the points indicated whether the mutations were associated with higher or lower disease risk, while their size conveys the strength of association measured through coefficients. Statistical significance was ascertained using a logistic model in PLINK, with the conventional threshold set at $5 \times 10^{-8}$ (before adjustment of multiple comparisons, two-sided $P$-value). **B** A scatter plot illustrating the convergence of GWAS signals ($P < 5 \times 10^{-8}$, two-sided, before adjustment of multiple comparisons) across identified common genes. The $x$-axis labels the phenotypes (IMDs), while the $y$-axis presents the coefficient for each association test. **C** The pleiotropic impacts of detected common protein-coding genes on various IMDs. The shape of the point denotes if the gene-disease link is novel, previously identified, or a replication of past findings. IMD immune-mediated disease, UKB the United Kingdom Biobank, MAF minor allele frequency, GWAS genome-wide association study, AD atopic dermatitis, Lichen Lichen planus, SLE systemic lupus erythematosus, Celiac Celiac disease, Crohn Crohn's disease, UC ulcerative colitis, T1D diabetes mellitus (Type I), Graves Graves' disease, HPT autoimmune hypothyroidism, SD sarcoidosis, AS ankylosing spondylitis, NV necrotizing vasculopathies, PR polymyalgia rheumatica, PST psoriatic and enteropathic arthropathies, Sicca Sicca syndrome (Sjogren's syndrome), MS multiple sclerosis, Behcet Behcet's disease, Hepatitis autoimmune hepatitis, ID immunodeficiency with predominantly antibody defects.

The FinnGen dataset supported 69 of the 115 associations under the threshold of $1.43 \times 10^{-3}$ (Supplementary Data 6). For 51 variants that were directly assessed without the need for neighboring substitutes, 42 (82.4%) were corroborated within the FinnGen dataset. For instance, *CFB* was observed to alleviate the risk associated with celiac disease (OR$_{UKB}$: 0.46, OR$_{FinGenn}$: 0.59) while *RNF5* heightened the risk for celiac disease (OR$_{UKB}$: 3.41, OR$_{FinGenn}$: 3.74). The results from the Asian population replicated 16 associations (including *BTNL2*-asthma), and the Black population replicated 7 associations (including *RNF5*-celiac disease; Supplementary Data 9).

We further ascertained the convergence between evidence from common WES variants and GWAS signals (i.e., the consistency in their identification of the same genes). Since GWAS typically implicates variants rather than specific genes, we mapped trait-linked SNPs to their candidate effector genes using refGene ("Methods" section). Of the 115 gene-disease associations that achieved exome-wide significance, 42 have been identified in previous GWAS studies. Before clumping, a total of 74 gene-disease associations also recognized significant GWAS signals. Notably, genes identified by common variants for conditions such as vitiligo, necrotizing vasculopathies (NV), Lichen planus, and Graves' disease exhibited 100% converge with those identified by GWAS (Fig. 3B and Supplementary Data 10). This suggested that these GWAS signals might be potentially influenced by independent WES variants (clumped by PLINK) in the proximal region.

Notably, many individual genes demonstrated associations with multiple IMDs, as depicted in Fig. 3C. In a demonstrative example, *BTNL2* exhibited associations with well-established IMDs phenotypes (asthma and Behcet's disease) and novel ones (celiac disease and ulcerative colitis [UC]). We further detected unique genetic overlap patterns across diseases. For example, *TYK2* exerted protective effects on both autoimmune hypothyroidism (HPT) and psoriasis.

## Genetic correlations among IMDs and their pleiotropic implications

Enlightened by the overlap of genes across IMDs revealed by exome-wide association analyses (Fig. 3C), we then systematically assessed the genetic correlations between IMDs. First, we estimated the genetic heritability of the IMDs based on rare variants using burden heritability regression (BHR)[37]. The contribution of the newly identified variants to heritability was estimated for 30 IMDs, with $h^2$ ranging from 0.2% to 22% (Fig. 4A and Supplementary Data 11). Among the IMDs, in terms of heritability, Behcet's disease (22.0%) ranked first, followed by allergic purpura (AP, 19.5%) and narcolepsy (18.7%). Ultra-rare pLOF variants accounted for the largest heritability contribution. Then, the genetic correlations across all pairwise combinations of IMDs were estimated[37]. A total of 395 BHR correlations between the disease pairs were identified (Fig. 4B and Supplementary Data 12), revealing the existence of widespread genetic correlations across IMDs, among which 181 were negative, and 214 were positive. The strongest association pairs were AD-gout ($R = 0.82$), and psoriasis-asthma ($R = -0.92$). These findings suggest that IMDs might share the mechanisms for their genetic liability.

Having revealed the shared genetic architecture among the genes and diseases, we wondered the pleiotropic effects of the identified

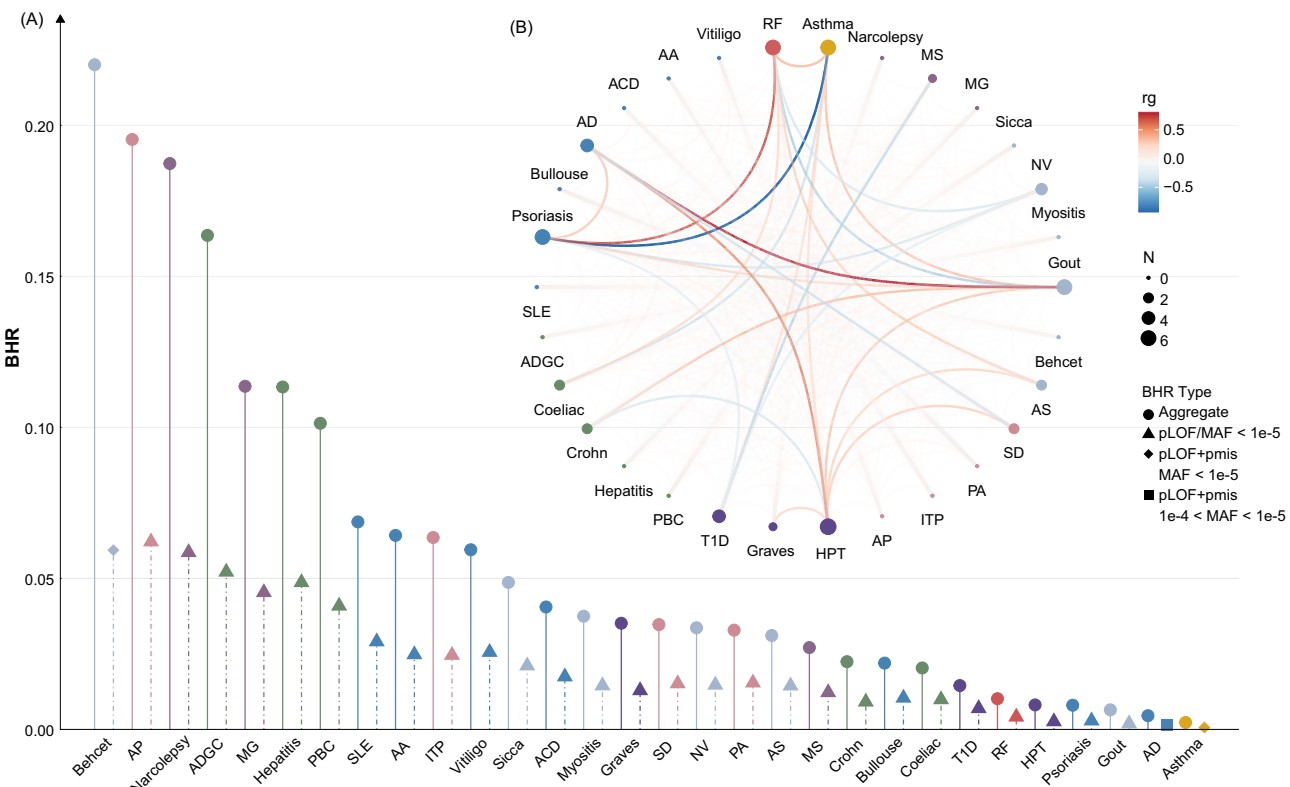

**Fig. 4 | Burden heritability and genetic correlations of IMDs. A** The burden heritability of IMDs calculated by burden heritability regression ("Methods" section). The *x*-axis indicates the specific IMDs and the *y*-axis indicates the heritability based on rare variants. The graph showcases the aggregate heritability for each disease and highlights the most impactful category of heritability for each phenotype. **B** Significant genetic correlations between the 30 IMDs with identified rare variants. Only substantial pairwise correlations (with a correlation coefficient, *rg* > 0.3) are emphasized, with weaker correlations appearing nearly transparent. The size of each node (represented by circles) indicates the number of significant correlations associated with a particular phenotype. The intensity of the line color between nodes conveys the strength and direction of the correlation coefficient.

IMD immune-mediated disease, BHR burden heritability regression, pLOF predicted loss-of-function, MAF minor allele frequency, pmis predicted deleterious missense, RF rheumatic fever, MS multiple sclerosis, MG myasthenia gravis, Sicca Sicca syndrome (Sjogren's syndrome), NV necrotizing vasculopathies, Behcet Behcet's disease, AS ankylosing spondylitis, SD sarcoidosis, PA pernicious anemia, ITP idiopathic thrombocytopenic purpura, AP allergic purpura, HPT autoimmune hypothyroidism, Graves Graves' disease, T1D diabetes mellitus (Type I), PBC primary biliary cirrhosis, Hepatitis autoimmune hepatitis, Crohn Crohn's disease, Celiac Celiac disease, ADGC allergic and dietetic gastro-enteritis and colitis, SLE systemic lupus erythematosus, Bullouse Bullouse disorders, AD atopic dermatitis, ACD allergic contact dermatitis, AA alopecia areata.

coding variants. To comprehensively assess this, we performed Phe-WAS across a total of 175 clinical outcomes identified by ICD-10 codes[38] in the UKB. For genes identified by rare variants, we identified 78 gene-disease associations, involving 10 disease classes under the FDR-corrected $P < 0.05$ (Supplementary Data 13 and Supplementary Fig. 2a). For example, *TET2* (idiopathic thrombocytopenic purpura [ITP]-related gene) was associated with Hodgkin's lymphoma, leukemia, pneumonia, and purpura. For common variants, we identified 1820 associations, involving 12 disease classes (Supplementary Data 14 and Supplementary Fig. 2b). For instance, *NOTCH4*, a highly heterogeneous gene and also the target for immune checkpoint inhibitor therapy[39], was revealed to be associated with multiple IMDs (celiac disease, Graves' disease, HPT, psoriasis and scleroderma).

**Protein-coding variants confer substantial risks for IMDs by time-to-event analysis**
To further characterize the disease relevance and clinical significance of the identified genes, we sought to quantify longitudinal disease risks for putatively pathogenic variations or annotated genes using Cox-proportional hazard ratio regression (Supplementary Data 15). 48 of 92 significant genes (52.2%) in rare variant analysis were revealed to be significantly related with the risks of corresponding diseases under FDR-corrected $P < 0.05$ (Supplementary Fig. 3). The median hazard

ratio (HR) for risk mutations stood at 5.71 (interquartile range [IQR]: 3.12–10.70), and that for protective mutations was 0.75. 32 out of 73 common variant genes (43.8%) significantly conferred altered risks of diseases, involving 42 gene-disease associations. The corresponding median HR was 1.16 (1.09–1.34) for risk mutations, and 0.96 (0.91–0.97) for protective mutations. As expected, the HRs for disease risk were generally higher in rare variants than in common variants. Of note, genetic evidence was consistent between time-to-event analyses and WES analyses. For instance, survival analysis also revealed that *TET2* (OR$_{burden}$: 1.17, $P_{SKAT} = 1.49 \times 10^{-9}$) increased the susceptibility to ITP (HR: 12.75, $P = 6.12 \times 10^{-14}$).

**Protein expression and druggability assessment**
The intergroup differences in gene expression between mutation carriers and non-carriers revealed that among 16 proteins available, a list of 12 proteins was found to be significantly altered (Fig. 5A and Supplementary Data 16). For example, significantly increased expression of LTA ($P = 2.20 \times 10^{-16}$) in celiac disease, IL18BP ($P = 3.28 \times 10^{-10}$) in Crohn's disease, and DDR1 ($P = 2.20 \times 10^{-16}$) in Graves' disease were observed. We subsequently utilized the MR approach to investigate the causalities, with the IVW method as the main model (Fig. 5B and Supplementary Data 17). Of the 12 altered protein, 5 protein-disease causalities were supported by MR, i.e., LTA-celiac disease

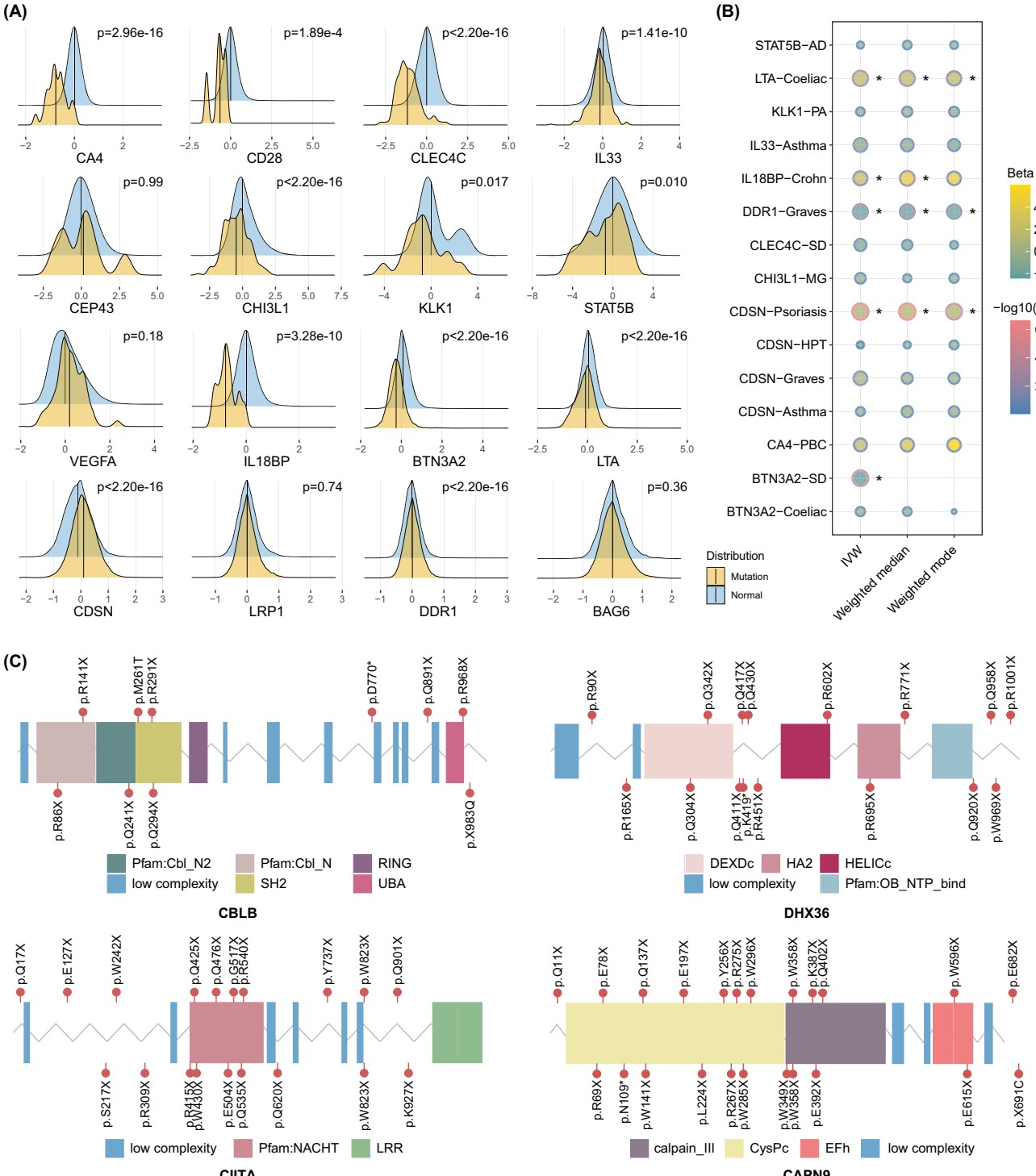

**Fig. 5 | Protein expression alterations and corresponding druggability. A** A series of density plots illustrating the differences in protein expression levels between mutation carriers and non-carriers. *P*-values are two-sided, adjusted by the false discovery rate. The black lines in each plot indicate the median protein expression level for each group. It's noteworthy that protein expression data is available for only 16 of the identified genes. **B** MR analysis to discern the causal link between protein expression levels and IMDs. GWAS analyses were first performed on protein expression, followed by the selection of SNPs from GWAS as instrumental variables. The point's edge color represents the negative logarithm of the FDR *P*-value, whereas the interior color stands for the coefficient. IVW was selected as the prior method for MR. **C** Gene plots displaying the protein-coding variants that contribute to the amino-acid signals for four protein entities (CBLB, DHX3, CIITA, and CAPN9). The protein domains and missense-constrained regions of the gene are also labeled. IMD immune-mediated disease, GWAS genome-wide association study, SNP single nucleotide polymorphism, FDR false discovery rate, MR Mendelian randomization, IVW inverse-variance weighted, AD atopic dermatitis, Celiac Celiac disease, PA pernicious anemia, Crohn Crohn's disease, Graves Graves' disease, SD sarcoidosis, MG myasthenia gravis, HPT autoimmune hypothyroidism, PBC primary biliary cirrhosis.

($P_{IVW} = 1.29 \times 10^{-3}$), IL18BP-Crohn's disease ($P_{IVW} = 4.04 \times 10^{-2}$), DDR1-Graves' disease ($P_{IVW} = 3.37 \times 10^{-3}$), CDSN-psoriasis ($P_{IVW} = 4.02 \times 10^{-7}$), and BTN3A2-SD ($P_{IVW} = 3.93 \times 10^{-5}$).

By annotating potential amino-acid alterations, we found that after gene mutations, a significant proportion of amino acids transited to terminators, as depicted in Fig. 5C (four protein entities: CBLB, DHX3, CIITA, and CAPN9) and Supplementary Data 18 (the rest proteins). This suggested that these mutations are likely to alter protein expression. For genes without available protein data, we employed proteomic-wide assessment to investigate whether their mutations altered the expression of other proteins. Operating under a statistical threshold of FDR-corrected $P < 0.05$, we discerned 80 associations between rare mutations (Supplementary Fig. 4a and Supplementary Data 19) and protein expression, and an extensive 14,592 associations for common mutations (Supplementary Fig. 4b and Supplementary Data 20). Remarkably, genes linked to a specific IMD phenotype could modulate proteins associated with other IMDs. For instance, CHI3L1 was influenced by both *PHACTR1* and *ZNF311*, CDSN by *DXO* and *PSMB9*, and LTA by *GSTM5*, *PSMB9*, and *ENSG00000244255*. Among them, *PSMB9* had 22 significant associations, predominantly with immune-related proteins, especially IL10 (Supplementary Fig. 5). Within the spectrum of the associations between common mutations and proteins, the most 25 pronounced associations (down-regulating) involved the protein of MICA/MICB, among which its association with *C6orf15* was most significant (OR: 0.21, $P < 1.00 \times 10^{-350}$). The shared regulatory and interaction network between different proteins (genes) suggests a high genetic overlap among these diseases, which provides significant implications for the development of drug repurposing.

Finally, we explored the druggability of the genes by querying the GeneCards with its associated DrugBank, HMDB, and Tocris databases. We found that 87 of 164 (53.0%) identified genes are druggable, including 4 causal genes (*LTA, DDR1, CDSN*, and *IL18BP*) supported by MR (Supplementary Data 21 and 22). They have been widely used to develop immunosuppressant or immunomodulators, including *LTA* (for Etanercept), *DDR1* (for Fostamatinib, Imatinib, Nilotinib, and Pralsetinib), and *CDSN* (for Carboplatin, and Gemcitabine).

### Biological insights into the identified gene-disease associations

To refine our understanding of how genetic variations confer risk for IMDs, we performed a series of bioinformatic analyses. First, we delved into the linkage with a series of biological indicators via PheWAS. The identified genes were significantly related with a range of biochemistry (especially, cholesterol-related indexes), inflammatory (white blood cell and neutrophil), spirometry (FEV1-related indexes), brain MRI (insula and posterior cingulate), and cognition (fluid intelligence and pair matching) traits (Fig. 6A, B and Supplementary Data 23 and 24). For instance, *LTA* was associated with a list of inflammatory (white blood cell, lymphocyte, neutrophil, and monocyte counts), and lipid-related (total cholesterol, triglycerides, and LDL cholesterol) indicators, highlighting the biological relevance with clinical celiac disease. Next, we adopted protein-protein interaction (PPI) analysis to construct an interaction network containing 164 genes and their disease-associated pathway clusters (Fig. 6C, Supplementary Fig. 6, and Supplementary Data 25). Generally, three cluster enrichment was observed under the FDR $P < 0.05$, basically covering the major functional pathways involved in the IMDs, supporting the biological validity of the genetic associations. Specifically, cluster 1 mainly involves immune-related pathways, cluster 2 is related to the urate metabolic process, and cluster 3 was of antigen processing and presentation. Besides focusing on the specific biological pathways, the target tissue and cell types should also be noted for the precision treatment of IMDs. We next characterized the specific cell types where the identified genes showed altered expression using single-cell RNA sequencing (scRNA-seq) data, which revealed diverse expression levels across varying cell and tissue types for a multitude of genes (Fig. 6D and Supplementary

Fig. 7). For instance, *LRP1* (asthma-related gene) is mainly expressed on neutrophils in the blood, but mainly expressed on macrophages and fibroblasts in the bladder. *TET2* (ITP-related gene) is widely expressed on various inflammatory cells (neutrophils, monocytes, and NK cells) in the blood, but is mainly expressed on macrophages in the bladder.

## Discussion

Here, we conducted a large-scale and comprehensive WES study of protein-coding variants for IMDs from 350,770 UKB individuals. We implicated 162 significant risk genes for 35 IMDs across the protein-coding allelic frequency spectrum, among which 124 were novel. Several genes showed independent convergence of rare and common variants evidence, reinforcing their vital value in the corresponding diseases. Longitudinal time-to-event analysis revealed that most of the observed genes (52.2%) significantly influenced disease onset at the population level. MR analysis further provided the causal evidence of 5 gene-disease associations, among which 4 were approved drug targets. By focusing on coding variants, we found mutations for specific IMDs not only significantly affected the protein expression in these IMDs, but also influenced the protein expression in other IMDs, revealing the shared network of mutual regulatory mechanisms across IMDs. Functional annotation provided biological insights into the gene-phenotype linkages by demonstrating the involvement of immune and metabolic pathways in these IMDs.

Traditional GWAS signals were not driven by the loss-of-function[40]. However, protein-coding variants have demonstrably greater translational potential, given their ability to interpret the functional impacts[41] and more clear effects on disease pathogenesis[42]. Our WES is the most comprehensive one to date to identify protein-coding genetic mutations across 40 IMDs. First, the congruence between our findings and previous research underscores the rigor and robustness of our study. 38 of the 164 genes through protein-coding variants have been identified in previous studies. For instance, we replicated the finding of *IFIH1*—the gene playing an important role in type I interferon production and signaling—are associated with psoriasis[43]. The proteome analysis further revealed the association between *IFIH1* and the lower blood level of protein IFNL1. IFNL1 was recently reported to be involved in the expression and production of all IFN-$\lambda$[44], and might be effective in the pathogenesis of psoriasis by regulating the biological potential of IFN-$\lambda$ signaling[45]. *SLC22A11* was previously reported to be associated with the risk of gout[46]. Through collapsing analysis in combination with conditional analysis adjusting the nearby common variants, we replicated this gene for the risk of gout, as well as supported by the FinnGen cohort. Our PheWAS analysis further revealed the association between *SLC22A11* and the blood levels of uric acid and Cystatin C. Second, we deciphered 124 novel putative pathogenic genes spanning across 35 IMDs. *TET2* was a novel gene for ITP through protein-coding variants, which was solid in longitudinal association analyses. Our proteome analysis revealed the association between *TET2* and lower blood levels of heme oxygenase 1 (HMOX1), which is involved in erythroid differentiation. We further found leukemia, purpura, and a series of blood-related parameters to be associated with this gene. All of the above analyses suggest the possibility of *TET2*, previously been developed for ascorbic acid, might be a novel drug target for ITP.

Among gene-level signals for which an individual variant also achieved significance, we highlighted the examples where both gene-based and single-variant-based genes contributed to disease burden. For instance, *FLG* was found in both gene-based (OR = 1.01) and single-variant-based (OR = 1.22) analyses for asthma. Time-to-event analysis revealed that *FLG* was correlated with a higher risk of asthma, with HRs ranging from 1.06 to 1.12. The consistency of exome-wide gene-based and single-variant-based associations, as well as the longitudinal disease risk at the population level, which also translates to an independent exome-sequenced cohort (FinnGen), increases the confidence

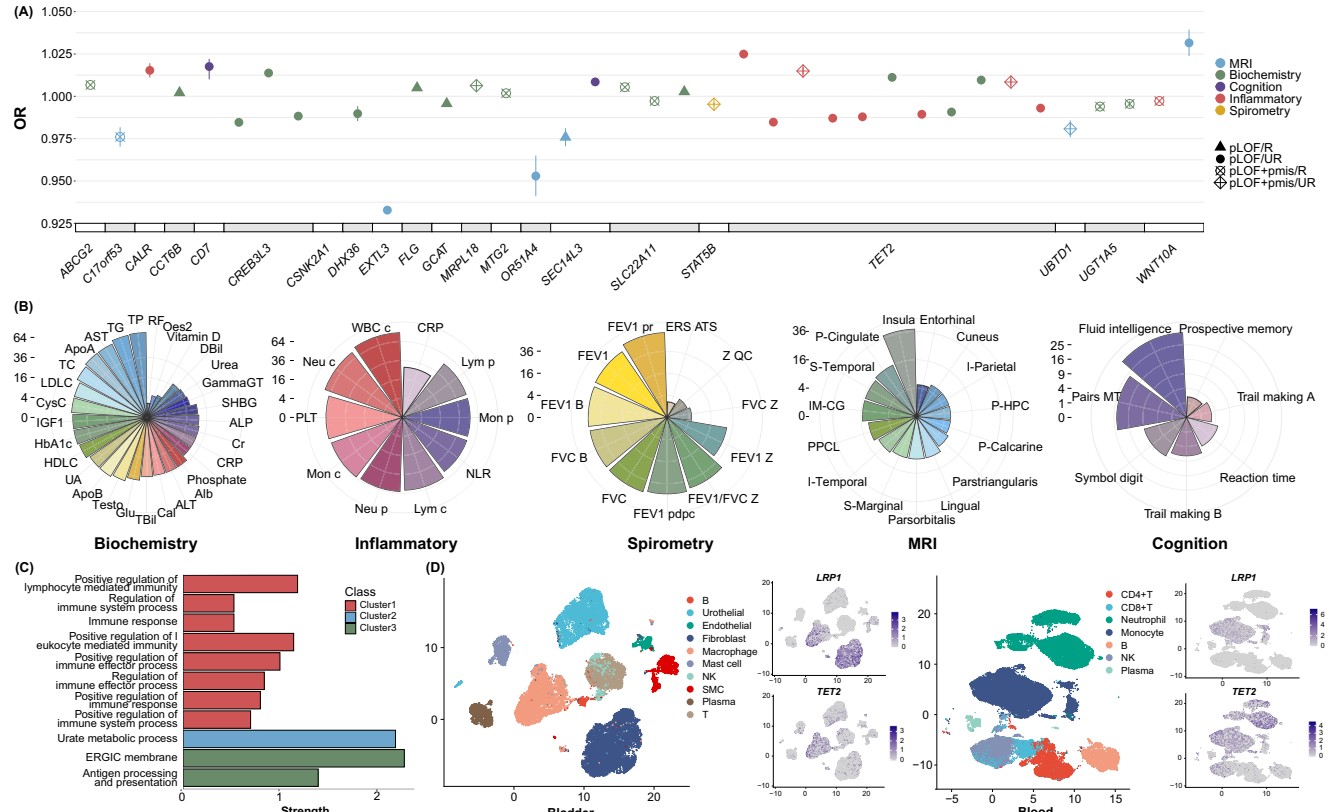

**Fig. 6 | Biological insights from multi-omics analysis.** All *p*-values are two-sided, false discovery rate corrected. **A** Significant associations between rare IMDs genes and biological indicators. The color and shape of the points indicate the class of the associated biological indicators and the class of mutations. **B** The number of significant associations between common IMDs genes and biological indicators. For MRI, only the regions with the top ten significant associations were displayed. **C** Significant pathway enrichment of PPI clusters in KEGG, GO, and Reactome. **D** Uniform Manifold Approximation and Projection of scRNA-seq data of blood, bone marrow, bladder, and kidney for *TET2* and *LRP1*. IMD immune-mediated disease, PC principal component, OR odds ratio, MRI magnetic resonance imaging, scRNA single-cell RNA, KEGG Kyoto Encyclopedia of Genes and Genomes, GO gene ontology, pLOF predicted loss-of-function, UR ultra-rare, R rare, pmis predicted deleterious missense, TP total protein, RF rheumatoid Factor, DBil direct bilirubin, GammaGT gamma glutamyltransferase, SHBG sex hormone-binding globulin, ALP alkaline phosphatase, Cr creatinine, CRP C-reactive protein, Alb albumin, ALT alanine aminotransferase, Ca calcium, TBil total bilirubin, Glu glucose, Testo testosterone, ApoB apolipoprotein B, UA urate, HDLC Hdl cholesterol, HbA1c glycated

hemoglobin, CysC cystatin C, LDLC Ldl direct, TC total cholesterol, ApoA apolipoprotein A, AST aspartate aminotransferase, TG triglycerides, WBC c white blood cell count, Lym p lymphocyte percentage, Mon p monocyte percentage, NLR neutrophil lymphocyte ratio, Lym c lymphocyte count, Neu p neutrophil percentage, Mon c monocyte count, PLT platelet, Neu c neutrophil count, NLR neutrophil count/lymphocyte count, FEV1 forced expiratory volume in 1-second (Fev1), FVC Z forced vital capacity (Fvc) Z-score, FEV1_FVC_ratio_Z Fev1/ Fvc ratio Z-score, ERS_ATS reproduciblity of spirometry measurement using ERS/ATS criteria, Z_QC spirometry QC measure, FEV1_pdpc forced expiratory volume in 1-second (Fev1), predicted percentage, FVC forced vital capacity (Fvc), FVC_B forced vital capacity (Fvc), best measure, FEV1_B forced expiratory volume in 1-second (Fev1), best measure, FEV1_pr forced expiratory volume in 1-second (Fev1), predicted, Pairs_MT pairs matching, P-cingulate posterior cingulate, IM-CG isthmuscingulate, S-Temporal superiortemporal, PPCL parsopercularis, I-Temporal inferiortemporal, S-Marginal supramarginal, I-Parietal inferiorparietal, P-HPC Parahippocampal, P-Calcarian Pericalcarine.

that our gene findings are indeed biologically relevant. It is also worth mentioning that the majority of rare mutations are disease-specific (91 of 93 gene-disease associations), while many common mutations are shared across IMDs (around 25%). Previous studies have also revealed the overlap of common mutations, for example, inflammatory bowel disease (IBD) shares their genetics not only with dermatological immune-mediated disorders but also with autoimmune endocrine disorders such as Type I Diabetes mellitus (T1D)[47]. Despite the shared genetic overlap among the IMDs, we further ascertained that many shared genes exhibit opposite effects. For example, mutations in *CFB* potentially confer protection against celiac disease but increase susceptibility to ankylosing spondylitis (AS) and UC. Previous research also observed such discordances (a shared locus for which the same haplotype increases risk for one disease but is protective for another), with a discordance rate of 14% between IBD, AS, and celiac disease in a sample of 416 instances[48]. The specificity between the IMDs was also reflected in that many genes appear to be more restricted to particular cellular contexts, which is consistent with one previous report[49].

For example, there is an over-representation of celiac disease loci expressed selectively in monocytes[50]; asthma-associated loci are preferentially expressed in CD4+ T cells[51]. Collectively, these empirical observations underscore the inherent heterogeneity concerning the influence of genetic mutations on IMDs phenotypic expressions. Future research should also focus on these disease-specific factors, as they may yield important clues to the disease mechanisms and may provide avenues for prevention and treatment.

The development of biological therapies targeting specific inflammatory proteins has transformed the clinical management of IMDs[52]. First, we pinpointed that the expression level of several genes significantly altered between the normal population and patients with the specific IMDs, i.e., decreased LTA protein levels in patients of celiac disease, highlighting the protein-coding functions of these genes and their clinical readabilities. Next, our proteomic analysis showed that *LTA* could significantly affect the blood levels of two cancer-related proteins-MUC16[53] and KLK11[54]. Previous studies have pointed out the possible associations between celiac disease and neoplasms, especially

malignancies[55]. The above evidence reinforced the involvement of *LTA* in celiac disease. Furthermore, evidence from our analyses yielded clues to a gene's biological mechanisms and relevance to diseases. Our PheWAS showed that *LTA* was associated with a list of inflammatory (white blood cell, lymphocyte, neutrophil, and monocyte counts), and lipid-related (total cholesterol, triglycerides, and LDL cholesterol) indicators, suggesting that the biological relevance of the gene is consistent with the well-established genetic relationships of clinical phenotypes. Monitoring clinical mutation-related biomarkers might help differentiate etiologies and guide individualized treatments[9], as well as contribute to genetic therapy. However, more importantly, understanding whether the proteins are drivers of disease is of vital importance for the development of treatments[23]. To this end, we used MR to evaluate the causal contributions of proteins to different IMDs. MR revealed four proteins including LTA, which is highlighted above, and it has already been approved for developing immunomodulators (Abacavir, Etanercept, and Carbamazepine), which mainly participates in the regulation of cellular apoptosis critical in the development of IMDs[56]. It is interesting and meaningful to explore whether there are activators targeting *LTA* that might be developed for the treatment of specific IMDs. *CDSN* is another causal gene supported by the MR evidence. It is a gene of considerable heterogeneity, located in the major histocompatibility complex (MHC) class I region on chromosome 6, and encoded a protein found in corneodesmosomes[57]. It is highly polymorphic in the human population, and its variation is associated with T1D, polymyalgia rheumatica, psoriatic/enteropathic arthropathies, and UC in the present study. The proteomic analysis demonstrated that *DXO* (celiac disease- and T1D-related gene) and *PSMB9* (alopecia areata-related gene) significantly altered the blood expression of CDSN, hinting that there might be a network of mutual regulatory mechanisms between these autoimmune-related genes. Moreover, *CDSN* has been used in the development of immunosuppressants (Carboplatin and Gemcitabine), providing confirmation of the utility of this approach, and also highlighting new potential therapeutic targets.

The major strength of this study lies in not only the systematic investigations of protein-coding variants in a series of 40 IMDs, but also the identification of a list of novel genes and exploration of their biological relevance. By virtue of focusing on coding variants, the observed associations more often provide a direct causal link between variants in a gene and a specific IMD[58], having implications for identifying or validating drug targets. However, the results should be viewed in light of several limitations. First, the ethnicity is mainly of European descent, making generalizability to other ethnicities challenging. Though the sub-population analyses in Black and Asian partly supported our findings and further provided more gene labels, other larger ancestral groups are no doubt valuable and can provide further information. Second, the sample sizes of some specific diseases were too small, resulting in insufficient statistical power, which may have resulted in missing genes. As we can see from Supplementary Fig. 1, the actual *P*-values were lower than the expected *P*-values for some IMDs. Third, the UK Biobank population may reflect volunteer bias and survivor bias with a sample of healthier individuals than the general UK population[59], and therefore may show a lower frequency of putative pathogenic variants and lower penetrance. Fourth, the observed effect sizes for genes identified through collapsing analysis may be lower than those reported in prior research. This discrepancy can be attributed to SAIGE's methodological rigor in correcting inflated type I error rates in binary traits with imbalanced case-control ratios and rare variants, thereby yielding more precise and biologically credible beta estimates[16,17,32,60]. Consequently, compared to earlier approaches, SAIGE's estimated effect sizes are less susceptible to inflation. Last, the FinnGen was not a true replication cohort. We acknowledge that our primary focus was not framed as replication but rather aimed at

providing evidence to support our results. Altogether, this study comprehensively examines the contribution of protein-coding variants to the genetic architecture of a series of complex IMDs, demonstrating the potential for gene-based analyses in large sequenced biobank cohorts. Investigations of the functional relevance of protein-coding variants might improve our understanding of the pathobiology of IMDs, lead to improved identification of affected pathways, and eventually can pave the way for the development of personalized therapies.

## Methods

### Study participants and disease phenotypes

The UKB is a population-based study that enrolled more than 500 thousand participants aged 40–69 years at recruitment across the UK. In-depth phenotypic, health-related, genetic, and proteomic data were collected from a baseline assessment and subsequent follow-up visits. UKB has obtained ethics approval from the Research Ethics Committee (REC; approval number: 06/MRE08/65) and informed consent from all participants. For this study, we included a total of 350,700 participants with available WES and clinical data after quality control (QC) under project application number 19542.

The IMDs phenotypes were ascertained and classified according to the in- and out-patient International Classification of Disease, Ninth Revision (ICD-9) and Tenth Revision (ICD-10) codes[61] and Read codes[62–66]. Data sources included the UKB health outcome records' first occurrences of health outcomes (Category 1712), hospital admission data (Field ID 41270, 41271), and primary care (Category 3000). The diagnose codes for IMDs were provided in Supplementary Data 26. We defined participants with a specific IMD as cases, otherwise as controls. Case status was determined at the last follow-up.

### Whole-exome sequencing and quality control

WES data were processed using the Regeneron Genetics Center (RGC) SPB pipeline[14]. Briefly, the Genomic DNA was extracted from blood samples and transferred to the RGC, and stored in an automated sample biobank at −80 °C. Sequencing was performed by dual-indexed 75 × 75-bp paired-end reads using the IDT xGen v1 capture kit on the Illumina NovaSeq 6000 platform.

The OQFE WES pVCF files provided by the UKB (https://biobank.ctsu.ox.ac.uk/showcase/label.cgi?id=170) were used, which was aligned to the human reference genome GRCh38[67]. In addition to the standard QC that was performed centrally, we performed additional genotype-, variant- and sample-based QC procedures to ensure a high-quality dataset for analyses[16] (Supplementary Methods). In brief, we first conducted a genotype refinement on the preliminary genotype calls present within the pVCF files utilizing Hail. Multi-allelic sites were segregated to yield distinct bi-allelic representations. Any calls failing to meet the hard filtering criteria were removed. Then we performed variant-based QC by excluding variants that exhibited a call rate of less than 90%, deviated from Hardy–Weinberg equilibrium ($P < 1 \times 10^{-15}$), or were present within regions of low complexity. Finally, on the sample level, we excluded participants who had rescinded their consent, instances of sample duplications, incongruences between genetically inferred sex and self-reported gender, as well as those samples displaying values for Ti/Tv, Het/Hom, SNV/indel, and singleton counts that deviated from mean±8 standard deviations (8sds).

The relatedness between samples was defined using the kinship coefficient score by KING software. The kinship coefficient threshold at 0.0884, which indicated the 2nd relatedness, was used to define an unrelated sample. We restricted our main analysis to the European-descent genetic ethnic group (field ID 22006) and utilized high-quality variants to compute the PCs pertinent to ancestry. Detailed methodologies are elaborated upon in the Supplementary Methods.

## Variant annotation

Rare variants (MAF < 1%) were annotated using SnpEff[68] v5.1 against Ensembl Build 38. Gene regions were defined using Ensembl Release. Predicted loss-of-function (pLOF) variants were those annotated to cause frameshift insertion/deletion, splice acceptor, splice donor, stop gain, start loss, and stop loss. Predicted deleterious missense (pmis) variants were defined as those predicted consistently to be deleterious by 5 in silico prediction tools including SIFT[69], LRT[70], PolyPhen2 HDIV, and PolyPhen2 HVAR[71], and MutationTaster[72]. Common variants (MAF ≥ 1%) were annotated with ANNOVAR[73] using refGene (https://annovar.openbioinformatics.org/en/latest/user-guide/download/) as a reference panel. We also combined positional mapping with eQTL and Chromatin interaction mapping by Functional Mapping and Annotation of Genetic Associations (FUMA; https://fuma.ctglab.nl/) to find the genes that they regulate. Variants annotated as exons were included in the following analysis.

## Rare variant burden analysis

Gene-level collapsing analysis was executed for rare variants. It encompassed 8 variant criteria, which vary in terms of MAF ($<1 \times 10^{-5}$, $<1 \times 10^{-4}$, <0.001, and <0.01), and predicted consequences (pLOF and a composite of pLOF and pmis). To discern genes pertinent to IMDs, a generalized mixed-effects model was deployed. All association analyses incorporated covariates including sex, age, and the first 10 PCs as fixed effects, serving to attenuate confounders and mitigate potential population stratification. The sparse genetic relationship matrix, which was used for the random effect variance ratio estimation in the model, was constructed using high-quality variants with the recommended relative coefficient cutoff of 0.05. For each collapsed association, effect sizes and the associated P-values were ascertained through testing modalities including SKAT[74], burden[75], and SKAT-O[76], implemented in SAIGE-GENE+ v1.1.6.2[77], with SKAT reported in the main article. The conventional threshold for collapsing analysis ($P_{SKAT} < 2.5 \times 10^{-6}$) was applied to determinant significant genes. Additionally, this analysis was extended to the specific Asian and Black populations (Field ID: 21000).

## Case-control enrichment of rare variants across consequence categories

To offer a more direct clinically relevant perspective on mutations, we stratified variants into pLOF and four discrete pmis classifications by utilizing the variant effect predictor software[78]. Case-control enrichment was central to our approach, which offers a clearer lens into which mutation types are of utmost clinical relevance. Instead of employing burden tests via SAIGE, we derived the odds ratio (ORs) by drawing from both penetrance and prevalence. Compared to the traditional burden tests, this approach ensures ease of computation, thus enhancing its clinical practicality.

The pLOF was delineated as per the prior analysis. The pmis variants were characterized using the Rare Exome Variant Ensemble Learner (REVEL) score[79]. We established five deleteriousness thresholds by sequentially categorizing variants based on decreasing levels of predicted deleteriousness: pLOF, REVEL ≥ 75, 75 ≥ REVEL > 50, 50 ≥ REVEL > 25, and 25 ≥ REVEL > 0. Based on established methods, we computed the case-control enrichment across consequence categories[37]. Let NCase be the number of cases and NMcase be the number of minor alleles in cases. The MAF of variants in cases (AFCase) was calculated by:

$$AFCase = \frac{NMcase}{2 * NCase} \quad (1)$$

Let NCtrl be the number of controls and NMctrl be the number of minor alleles in controls. The MAF of variants in the total population (AF) was calculated by:

$$AF = \frac{NMcase + NMctrl}{2 * (NCase + NCtrl)} \quad (2)$$

We calculated the per-sd coefficients using AF, AFCase, and prevalence:

$$\beta_{pq} = \frac{2(AFCase - AF) * prevalence}{\sqrt{2AF(1 - AF) * prevalence * (1 - prevalence)}} \quad (3)$$

where prevalence equals to Ncase/(NCase + NCtrl).

Variant variances were calculated by:

$$twopq = 2 * AF * (1 - AF) \quad (4)$$

Finally, by combining (3) and (4), we converted beta per-sd (i.e., sqrt(variance explained)) to per-allele (i.e., in units of phenotypes):

$$\beta = \frac{\beta_{pq}}{\sqrt{twopq}} \quad (5)$$

## Single association analysis for common variants

For exon SNPs exhibiting an MAF of 1% or higher, a single association examination was conducted among the non-related European-descent cohort utilizing the computational tool PLINK2[36]. Lead SNPs were defined as independent significant SNPs with $r^2 \le 0.1$ within a 1 Mb window. Covariates encompassing age, sex, and the ten PCs were adjusted. The significance threshold was established at the conventional threshold for GWAS ($P < 5 \times 10^{-8}$).

## External replication in FinnGen

To validate our discerned associations in an independent cohort, we leveraged summary statistics from the FinnGen Consortium online results (version 8)[35]. The FinnGen study amalgamates genotypic information with national health registry data of Finnish citizens. The summary statistics were publicly online available (see "Data availability" section). Genotype-, sample- and variant-wise QC and filtering procedures can be found in previous studies. For the validation of rare mutations, coefficient values and P-values were initially calculated and then we obtained the results of gene-disease associations by selecting the strongest associations (i.e., lowest P-value) per gene. For common mutations, we obtained the coefficients and P-values for the according variant if available. If the variant was filtered in QC or not imputed, we utilized variants within 50 kb as a substitute. A Bonferroni-corrected threshold of $P < 1.43 \times 10^{-3}$ (0.05/35 IMDs) was considered to be supported. The precise diagnosis code, along with the case and control distribution for each phenotype were delineated in Supplementary Data 27.

## Genome-wide association study

Variants that did not deviate from HWE ($P > 1 \times 10^{-12}$), per variant missing rates 1%, and an imputation quality score >0.8 were selected. We carried out linear mixed model association analyses and adjusted for the genotyping array and 10 ancestry PCs to assess the associations between the traits and imputed genotype dosages under an additive genetic model by using BOLT-LMM v2.3. Details were addressed in Supplementary Methods.

## Heritability and proportion of variance-explained estimates

Burden heritability regression (BHR)[37] was applied to estimate the heritability explained by the rare variants using the R package BHR v0.1.0 (accessible at: https://github.com/ajaynadig/bhr). BHR regresses gene burden statistics on gene burden scores and estimates the

burden heritability from the regression slope. We computed per-allele effect sizes from case and control allele counts as the input to BHR, as recommended by the previous publication[37]. Our investigation focused on pLOF and pmis variants. The univariate burden heritability was separately computed across four distinct MAF intervals, namely: $(0, 10^{-5})$, $[10^{-5}, 10^{-4})$, $[10^{-4}, 10^{-3})$, and $[10^{-3}, 0.01)$ and then further aggregated into total heritage. Bivariate BHR was further performed to calculate the genetic correlations between phenotypes.

### Associations with clinical outcomes by Cox analysis
For rare mutations, we defined those carrying pLOF or deleterious pmis mutations in the identified gene as cases, otherwise as controls. For common mutations, the analysis was stratified by the number of identified mutations (0,1,2) that a subject carried. We tested the identified likely-causal variants for associations with corresponding disease outcomes. Time zero corresponded to the birth year of each individual, and follow-up time was subsequently calculated as years of birth to the date of first diagnosis, death, or the final date with accessible information from hospital admission, whichever came first. After FDR correction, $P < 0.05$ was identified as significant.

### Gene expression analysis
T-tests were used to identify the protein expression alterations in subjects carrying mutations and non-carriers. Mutation carriers were defined as carrying the specific identified variants in the common variant spectrum or carrying any of the pLOF or deleterious pmis variants in the rare variant spectrum. A density plot was crafted to illustrate the differential expression. It is imperative to note that the UKB furnished data pertaining to only 1900 proteins. As such, only 16 genes are complemented with corresponding protein expression data. The population was restricted to a European-descent ethnic group, and protein expression levels outside 5sds were excluded.

### Mendelian randomization analysis
We delved into an exploration of the potential causal relationships between protein expression alterations and the corresponding IMDs via a two-sample MR approach. This method leverages genetic variants as instrumental variables, establishing a causal linkage between exposures and outcomes. Adjusting for age, sex, and the first 10 PCs of ancestry, we computed the GWAS (Supplementary Methods) through PLINK2. For GWAS of IMDs, we leveraged summary statistics from the FinnGen cohort to avoid sample overlap.

We employed several analytical models: inverse-variance weighted (IVW), weighted median, simple median, and weighted mode—all of which were operationalized via the R package TwoSampleMR (https://mrcieu.github.io/TwoSampleMR/). We selected genetic instruments characterized by $P$-values less than $1 \times 10^{-5}$ and subsequently performed Steiger filtering, ensuring the veritable association of the instrumental variables with the exposure. We clumped SNPs by excluding those with $r^2 > 0.01$ and kb<5000 to avoid the potential linkage disequilibrium. Cochran's Q-statistic was used to evaluate the heterogeneity inherent in the IVW. Radial MR and leave-one-out analysis was leveraged to eliminate SNPs with significant heterogeneity until no heterogeneity was observed by Q-statistic. We further adopted MR-Egger to assess potential pleiotropy, which detected no sign of pleiotropy in our MR. Since no heterogeneity and pleiotropy effect were observed, we prior the results from the IVW method.

### Amino-acid signals and Proteomic-wide analysis
We annotated the amino-acid alterations by ANNOVAR, delineating the structural framework and domains based on the SMART database. We subsequently conducted a proteomic-wide analysis to assess the associations between the putative mutations and the proteomic data by employing linear regression. Our analysis data was restricted to the European-descent demographic, omitting observations that deviated beyond 5sds. Covariates, namely age, sex, and the first 10 PCs were incorporated. An FDR-corrected $P$-value threshold of less than 0.05 was established as the criterion for statistical significance.

### Phenome-wide association study
A PheWAS approach was used to determine the clinical measures and comorbidities associated with the identified rare/common genes and loci. We included a total of 153 biochemistry, inflammatory, spirometry, cognition, and MRI (brain and heart) traits (Supplementary Data 28). We also assessed the pleiotropic effects of IMDs-associated genes on a total of 187 clinical outcomes identified by ICD-10 codes[38] (Supplementary Data 29). Biological indicators or disease phenotypes were tested for associations with the identified genes or single variants. The regressions (SAIGE for rare mutations and logistic/linear regressions for common mutations) were adjusted for age, sex, and the first 10 PCs. The results were deemed significant at an FDR level of 5%.

### Protein-protein interaction analysis
The Search Tool for the Retrieval of Interacting Genes database (STRING) (Version 10.0, http://string-db.org) was used to predict the relationships among the screened genes. Based on experimental data, database entries, and co-expression, PPI node pairs with a score of combination > 0.4 (medium confidence) were considered to be significant. A machine learning method (K-means) was utilized to categorize gene clusters.

### Functional enrichment analysis
To unravel the underlying biological pathways within the identified genes, we employed Multi-marker Analysis of GenoMic Annotation (MAGMA)[80] within the FUMA platform[81] to probe three of the most robust biological annotation and pathway compendiums: Gene Ontology (GO)[80], Kyoto Encyclopedia of Genes and Genomes (KEGG)[82], and Reactome[83]. The top ten significantly enriched pathways of the total gene sets and significantly enriched pathways of each cluster from PPI were displayed. FDR-adjusted $P < 0.05$ was statistically significant.

### Single-cell expression
A large single-cell RNA sequencing dataset by Tabula Sapiens Consortium[84] was obtained to identify the expression level of the genes in each cell type. The Tabula Sapiens was generated with data from 15 human donors, comprising ~500,000 cells from 24 different organs or tissues[84]. The clustered and annotated scRNA-seq datasets of blood, bone marrow, bladder, and kidney were obtained, and the R package Seurat was used for analysis and visualization[85].

### Reporting summary
Further information on research design is available in the Nature Portfolio Reporting Summary linked to this article.

## Data availability
The individual-level data used in this study were accessed from the UKB database under accession code 19542 (https://www.ukbiobank.ac.uk/). The exome sequencing data are available under restricted access for the data restriction policy of the UKB cohort, but access can be obtained by following the application instructions at https://www.ukbiobank.ac.uk/enable-your-research/apply-for-access. FinnGen summary statistics are publicly accessible (http://r8.FinnGen.fi). The raw whole-exome sequencing data are protected and are not available due to data privacy laws. The processed whole-exome sequencing data are available at https://biobank.ndph.ox.ac.uk/showcase/label.cgi?id=170. The gene-level and single-variant association summary statistics of the exome association study have been made accessible through https://doi.org/10.6084/m9.figshare.25420873.

## Code availability

Open-source R package SAIGE-GENE+ v1.1.6.2 was used to run gene-based collapsing tests for rare variants and the code was available from the GitHub (https://github.com/saigegit/SAIGE). PLINK v1.9 (https://www.cog-genomics.org/plink/1.9/) and v2.0 (https://www.cog-genomics.org/plink/2.0/) were adopted to perform association tests for common variants, variant quality control and sample quality control. Hail is adopted to perform genotype quality control (https://hail.is/). KING 2.3.1 is adopted to identify duplicated samples (https://www.kingrelatedness.com/). Rare variants were annotated by SnpEff v5.1 (https://pcingola.github.io/SnpEff/se_introduction/), ensembl variant effect predictor v101.0 (https://www.ensembl.org/info/docs/tools/vep/), and common variants were annotated by ANNOVAR. The burden heritability regression and genetic associations were performed using BHR v0.1.0 (https://github.com/ajaynadig/bhr). Protein-protein interactions and pathway enrichment were performed by STRING v12.0 (https://cn.string-db.org/). Protein-protein interactions and pathway enrichment were performed by STRING v12.0 (https://cn.string-db.org/) and function annotation was performed by FUMA. The code of the main analysis and visualization of single-nucleus RNA-seq data was an adaptation of the R package Seurat version 4.0 and is available from https://satijalab.org/seurat/index.html. Mendelian randomization was performed by using the R package TwoSampleMR. The figures are generated using the 'ggplot2' and 'ggbreak' packages. Codes are available at https://github.com/Sirius-Yang/IMDs_WES[86].

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

## Acknowledgements

The authors gratefully thank all the participants and professionals contributing to the UKB. We also want to acknowledge the participants and investigators of the FinnGen study. This study was supported by grants from the Science and Technology Innovation 2030 Major Projects (2022ZD0211600), National Natural Science Foundation of China (82071201, 82071997), Shanghai Municipal Science and Technology Major Project (2018SHZDZX01), Research Start-up Fund of Huashan Hospital (2022QD002), Excellence 2025 Talent Cultivation Program at Fudan University (3030277001), Shanghai Talent Development Funding for The Project (2019074), Shanghai Rising-Star Program (21QA1408700), 111 Project (B18015), and ZHANGJIANG LAB, Tianqiao and Chrissy Chen Institute, the State Key Laboratory of Neurobiology and Frontiers Center for Brain Science of Ministry of Education, and Shanghai Center for Brain Science and Brain-Inspired Technology, Fudan University.

## Author contributions

L.Y., Y.N.O., and B.S.W. organized data and carried out the statistical analysis. Y.N.O. and L.Y. participated in the first draft of the manuscript. L.Y. and W.S.L. designed and drew the figures. Y.T.D., X.Y.H., Y.L.C., J.J.K., C.J.F., and Y.Z. participated in the revision of the manuscript. L.T., Q.D., J.F.F., W.C., and J.T.Y. participated in the study design, reviewing and editing the manuscript. All authors read and approved the final manuscript.

## Competing interests

The authors declare no competing interests.
