## [Peer Review File · Nature Communications]

Large-scale whole exome sequencing analyses identified protein-coding variants associated with immune-mediated diseases in 350770 adultsREVIEWER COMMENTS

Reviewer #1 (Remarks to the Author):

Review report

Title: Large-scale whole exome sequencing of immune-mediated diseases in 350770 adults

Corresponding Author: Jin-Tai Yu

In this paper the authors describe the results of very comprehensive analysis from whole exome sequencing material containing 350770 UK Biobank participants. The aim was to identify the putative variants, especially predicted loss-of-function and predicted deleterious missense, across immune-mediated diseases and elucidate their clinical impacts in relation to biochemical processes. The study identified 162 unique genes across 35 immune-mediated diseases with 124 being novel. The results show a lot of interesting and important new information of the genes and their rare and common, risk and protective SNP variants involved in immune-mediated diseases. The data showing that mutations influenced disease onset, causal evidence of gene-disease associations, shared protein expression in different immune-mediated diseases and validation analysis with an independent exome-sequenced cohort are very important information for the clinical and research community working with immune-mediated disease. For a researcher, the extensive data tables may provide a welcome possibility to evaluate the meaning of not well-known gene variants on specific immune-mediated diseases. In addition to aforementioned aspects I have some specific comments:

-Nowadays, instead of using Caucasian individuals, you should use European-descent. In my mind this kind of paper needs to describe more precisely the cohort used for the study, especially when we know that the frequencies of gene variants related to immune-mediated diseases varies between populations as is shown in this paper between study and validation cohorts.

- Please describe the control population.

-Page 4 (72): deleterious

-Page 6 (108): asthma

-Page 7(126): asthma

-Page 44: Fig1 miss the letters A, B, C, D

-Figures miss numbers

Reviewer #2 (Remarks to the Author):

Yang et al. uses data from the UK Biobank to perform gene-level association with immune associated diseases. Methodological choices in the identification of people with specific diseases, association analyses, and linking non-coding variants to the genes that they regulate limit this reviewer's enthusiasm.

Major concerns:

- The algorithms used to identify the immune associated diseases need to be more clearly presented and referenced for their effectiveness.
- The manuscript starts with a gene-based assessment. One of the findings presented as being consistent with prior literature is FLG with atopic dermatitis – but the reported effect size is much less than previous studies that focus on loss of function variants in FLG.
- ANNOVAR was published in 2010 and was a good choice for years. In 2024, using 500 kb windows to link DNA with genes is not appropriate. Methods to consider include eQTLs, ABC, and similar approaches that use functional genomic data to match DNA to the gene that they regulate.
- For the FinGen replication study, it will again be important to present clearly how this study is different than the FinGen flagship manuscript published with the newest data release: Kurki, M.I., Karjalainen, J., Palta, P. et al. FinnGen provides genetic insights from a well-phenotyped isolated population. Nature 613, 508–518 (2023). <https://doi.org/10.1038/s41586-022-05473-8>.
- The Replication studies were presented such that p-values less than 0.05 were considered replication. This level of replication can be expected by chance given the number of association

analyses. Permutation testing is needed to identify if the signals did truly replicate more than expected by chance.

- The UK Biobank has been used by countless studies that focus on immune-related diseases. It is critical for this manuscript to clearly identify how this study relates to, replicates, and builds upon at least some of these studies including by identifying methodological differences.

Minor

- It is unclear what is meant on line 73 "comprehensive and solid WES studies". Consider rewording.
- The term "Caucasian" is problematic and should be removed throughout. Consider following updated guidelines around ancestry and race when naming groups of people.
- The term "risky pattern" (e.g. line 107) should be replaced with more precise terminologies.

Reviewer #3 (Remarks to the Author):

The authors present a large, systematic study of both rare and common coding variant effects on 35 immune mediated diseases (IMDs). By doing so, they discover novel disease-gene relationships and also characterise pervasive pleiotropy present amongst IMDs and other traits and biomarkers.

Overall, I think the authors results and conclusions are sound. I have several minor comments that should help increase the clarity of the presented work:

1. The paper is poorly written and has many different grammatical mistakes. For example, L56-57, L59, L107, L140, L249, just to name a few. The authors should work with the editor to ensure that this is systematically addressed before publication.

2. It was unclear to me how $P < 0.02$ (FDR) was determined? What P-value threshold does this come to, and how was it deduced?

3. Are the authors correcting for the fact they have performed 35 exome-wide association analyses? For example in the Finngen analysis, a replication P-value of 0.05 is used but the author should present replication at $P < 0.05/\text{number_of_traits_analysed}$

4. Do the authors adjust for sex, age or BMI in the exome associations? IMDs are very sex specific both in terms of incidence and genetics, the authors should adjust for sex or ideally, carry out sex-specific analyses.

5. How do the authors intend to share their results with the community? Is there a website that allows researchers to query these results and download summary statistics. I insist for all reviews that both code (github) and summary level findings are made publicly available.

Responses to the Reviewers

The authors sincerely appreciate the critical reviews of the paper, and for the helpful way in which the reviewing editors put together a constructive list of suggestions for the revision of the paper. We have now revised the paper to carefully address all the points raised. Our responses below are preceded by “ - ”, and changes made to the paper are shown below within “...”, and in **red font** in the revised paper.

REVIEWER COMMENTS

Reviewer #1 (Remarks to the Author):

Review report

Title: Large-scale whole exome sequencing of immune-mediated diseases in 350770 adults

Corresponding Author: Jin-Tai Yu

In this paper the authors describe the results of very comprehensive analysis from whole exome sequencing material containing 350770 UK Biobank participants. The aim was to identify the putative variants, especially predicted loss-of-function and predicted deleterious missense, across immune-mediated diseases and elucidate their clinical impacts in relation to biochemical processes. The study identified 162 unique genes across 35 immune-mediated diseases with 124 being novel. The results show a lot of interesting and important new information of the genes and their rare and common, risk and protective SNP variants involved in immune-mediated diseases. The data showing that mutations influenced disease onset, causal evidence of gene-disease associations, shared protein expression in different immune-mediated diseases and validation analysis with an independent exome-sequenced cohort are very important information for the clinical and research community working with immune-mediated disease. For a researcher, the extensive data tables may provide a welcome possibility to evaluate the meaning of not well-known gene variants on specific immune-mediated diseases. In addition to aforementioned aspects I have some specific comments:

1. Nowadays, instead of using Caucasian individuals, you should use European-descent. In my mind this kind of paper needs to describe more precisely the cohort used for the study, especially when we know that the frequencies of gene variants related to immune-mediated diseases varies between populations as is shown in this paper between study and validation cohorts.

Response:

- Many thanks for your positive feedback towards our article. According to your suggestions and the updated guidelines around ancestry and race, we have changed the term “Caucasian” to “**European-descent**” in the revised manuscript.

2. Please describe the control population.

Response:

- Thanks for your rigorous concerns. We defined participants with IMDs as cases, otherwise as controls. Case status was determined at last follow-up. The IMDs phenotypes were ascertained and classified according to the in- and out-patient International Classification of Disease, Ninth Revision (ICD-9) and Tenth Revision (ICD-10) codes and Read codes.
- We have added some details in the revised Methods section, **Page 23, Line 472-474** “**We defined participants with IMDs as cases, otherwise as controls. Case status was determined at last follow-up.**”

3. Page 4 (72): deleterious

Response:

- Thanks for your rigorous consideration. We are sorry for the spelling mistakes and have corrected the wrong spelling to “**deleterious**”.

4. Page 6 (108): asthma

Response:

- We are sorry for the spelling mistakes and have corrected the wrong spelling to “**asthma**”.

5. Page 7(126): asthma

Response:

- We are sorry for the spelling mistakes and have corrected the wrong spelling to “**asthma**”.

6. Page 44: Fig1 miss the letters A, B, C, D

Response:

- Thanks for your rigorous consideration. We have added the letters A, B, C and D in the revised Figure 1 and the related Figure legends.

7. Figures miss numbers

Response:

- Thanks for pointing out this issue. We are sorry for that we didn't add the numerical order of the figures at the end of the document. We have added the “**Figure 1-Figure 6**” beneath the related figures. We are not sure whether we understand your comments correctly, please feel free to contact us if you have any other comments.

Reviewer #2 (Remarks to the Author):

Yang et al. uses data from the UK Biobank to perform gene-level association with immune associated diseases. Methodological choices in the identification of people with specific diseases, association analyses, and linking non-coding variants to the genes that they regulate limit this reviewer's enthusiasm.

Major concerns:

1. The algorithms used to identify the immune associated diseases need to be more clearly presented and referenced for their effectiveness.

Response:

- Thanks for your rigorous concerns. The IMDs phenotypes were ascertained and classified according to the in- and out-patient International Classification of Disease, Ninth Revision (ICD-9) and Tenth Revision (ICD-10)¹ codes and Read codes²⁻⁶. Data sources include the UKB health outcome records' first occurrences of health outcomes (Category 1712), hospital admission data (Field ID 41270, 41271) and primary care (Category 3000), while self-report cases were excluded. The diagnose codes for IMDs were provided in Supplementary Table 26.
- We have cited the references to support the algorithms sources. Please refer to the “Methods-Study participants and disease phenotypes” section, **Page 22-23, Line 467-474**.

References:

- 1 Yokoyama, J. S. *et al.* Association Between Genetic Traits for Immune-Mediated Diseases

- and Alzheimer Disease. *JAMA Neurol* **73**, 691-697, doi:10.1001/jamaneurol.2016.0150 (2016).
- 2 de Lusignan, S. *et al.* Atopic dermatitis and risk of autoimmune conditions: Population-based cohort study. *The Journal of allergy and clinical immunology* **150**, 709-713, doi:10.1016/j.jaci.2022.03.030 (2022).
- 3 Rafiq, M. *et al.* Allergic disease, corticosteroid use, and risk of Hodgkin lymphoma: A United Kingdom nationwide case-control study. *The Journal of allergy and clinical immunology* **145**, 868-876, doi:10.1016/j.jaci.2019.10.033 (2020).
- 4 Persson, M. S. M. *et al.* Validation study of bullous pemphigoid and pemphigus vulgaris recording in routinely collected electronic primary healthcare records in England. *BMJ Open* **10**, e035934, doi:10.1136/bmjopen-2019-035934 (2020).
- 5 Cipolletta, E. *et al.* Association Between Gout Flare and Subsequent Cardiovascular Events Among Patients With Gout. *Jama* **328**, 440-450, doi:10.1001/jama.2022.11390 (2022).
- 6 Schonmann, Y. *et al.* Inflammatory skin diseases and the risk of chronic kidney disease: population-based case-control and cohort analyses*. *British Journal of Dermatology* **185**, 772-780, doi:10.1111/bjd.20067 (2021).

2. The manuscript starts with a gene-based assessment. One of the findings presented as being consistent with prior literature is FLG with atopic dermatitis – but the reported effect size is much less than previous studies that focus on loss of function variants in FLG.

Response:

- We greatly appreciate your astute observations and valuable insights. We acknowledge that there are two main reasons for the difference in effect sizes between our study and previous studies: the population characteristics of UKB and the intrinsic properties of analytical models we used.

- We have provided the potential reasons in the Discussion section, **Page 21, Line 439-446** (“Third, the UK Biobank population may reflect volunteer bias and survivor bias with a sample of healthier individuals than the general UK population², and therefore may show lower frequency of putative pathogenic variants and lower penetrance. Fourth, some effect sizes for identified genes from collapsing analysis are lower than expected from previous studies. We hypothesized that it was primarily due to usage of saddle point-approximation-corrected logistic mixed-model approach implemented in SAIGE-GENE+ software, which might yield

slightly conservative effect estimates, particularly when assessing significance for binary traits with imbalanced case-control ratios².”)

References:

1 Schoeler, T. *et al.* Participation bias in the UK Biobank distorts genetic associations and downstream analyses. *Nature human behaviour* **7**, 1216-1227, doi:10.1038/s41562-023-01579-9 (2023).

2 Jurgens, S. J. *et al.* Analysis of rare genetic variation underlying cardiometabolic diseases and traits among 200,000 individuals in the UK Biobank. *Nat Genet* **54**, 240-250, doi:10.1038/s41588-021-01011-w (2022).

3. ANNOVAR was published in 2010 and was a good choice for years. In 2024, using 500 kb windows to link DNA with genes is not appropriate. Methods to consider include eQTLs, ABC, and similar approaches that use functional genomic data to match DNA to the gene that they regulate.

Response:

- Thanks for pointing out this issue. According to your suggestions, in order to match DNA to the gene they regulate, we combined positional mapping with eQTL and Chromatin interaction mapping by FUMA (<https://fuma.ctglab.nl/>). A total of 489 genes were mapped by position or eQTL or Chromatin interaction mapping and 109 genes (94.8% overlapped with ANNOVAR) were consistently mapped. Therefore, we believe that the results of these two annotation strategies (positional annotation and functional annotation) are not very different.
- We have added this issue in Supplementary Table 7 and revised manuscript, **Page 9, Lines 167-173: Results** – “Different from positional annotation conducted by ANNOVAR, we also combined positional mapping with eQTL and Chromatin interaction mapping by FUMA (<https://fuma.ctglab.nl/>) to find the gene that they regulate. A total of 489 genes were mapped by positional or eQTL or Chromatin interaction mapping and 109 genes were consistently mapped (94.8% overlapped with ANNOVAR; Supplementary Table 7-8).”.

4. For the FinGen replication study, it will again be important to present clearly how this study

is different than the FinGen flagship manuscript published with the newest data release: Kurki, M.I., Karjalainen, J., Palta, P. et al. FinnGen provides genetic insights from a well-phenotyped isolated population. *Nature* 613, 508–518 (2023). <https://doi.org/10.1038/s41586-022-05473-8>.

Response:

- Thanks for your rigorous concerns. We first apologize for our ambiguous expression. In our study, to validate our discerned associations in an independent cohort, we leveraged summary statistics from the FinnGen Consortium online results (version 8)¹. The summary statistics were publicly online available (<https://r8.finngen.fi/>). We searched for the coefficient value and p-value for each gene identified in our gene-disease associations and selected the strongest associations (i.e., lowest p value) per gene. And for common mutations, we obtained the coefficient and p-value for the according variant if available.
- We now acknowledge that our primary focus was not framed as a replication but rather aimed at providing evidence to support our results. We have clarified this in the Revised Manuscript and deleted redundant information (**Page 7, Lines 129-133: Results-** “In order to validate the gene-based associations in UKB, we searched from Kurki et al’ s summary statistics analyzed from FinGenn dataset.”. **Page 27-28, Lines 573-598: Methods** - “To validate our discerned associations in an independent cohort, we leveraged **summary statistics** from the FinnGen Consortium **online results (version 8)**¹. ... **The summary statistics were publicly online available (see data availability)**. Genotype-, sample- and variant-wise quality control and filtering procedures can be found in previous study¹. ...**A Bonferroni-corrected threshold of $P < 1.43 \times 10^{-3}$ (0.05/35 IMDs)** was considered to be supported. The precise diagnosis code, along with the case and control distribution for each phenotype, is delineated in Supplementary Table 27.”).

References:

1 Kurki, M.I., Karjalainen, J., Palta, P. et al. FinnGen provides genetic insights from a well-phenotyped isolated population. *Nature* 613, 508–518 (2023).

5. The Replication studies were presented such that p-values less than 0.05 were considered replication. This level of replication can be expected by chance given the number of association analyses. Permutation testing is needed to identify if the signals did truly replicate more than expected by chance.

Response:

- Thank you for your rigorous consideration. First, to be clear again, we leveraged summary statistics from the FinnGen Consortium (version 8)¹ to validate our discerned associations. Unfortunately, the permutation testing can't be conducted with the summary-level statistics.
- Next, to exclude the contingency of validation, we applied a more stringent threshold of Bonferroni-corrected for the number of traits analyzed (0.05/35). We have clarified this in the Revised Manuscript (**Page 7, Lines 134-139: Results**—"Searches yielded 13 associations (19% replicated) of nominal significance (P value $<1.43\times 10^{-3}$; Supplementary Table 2). Notably, *FLG* for AD (OR: 2.09, $P=6.54\times 10^{-53}$), *ETV7* for Primary biliary cirrhosis (PBC, 1.98, $P=2.21\times 10^{-4}$), and *ABCG2* for gout (1.69, $P=2.73\times 10^{-76}$) emerged with the most pronounced coefficients...", **Page 9, Lines 174-175: Results**—"The FinnGen dataset supported 69 of the 115 associations under the threshold of 1.43×10^{-3} (Supplementary Table 6)." and **Page 28, Lines 595-596: Methods**—"A Bonferroni-corrected threshold of $P<1.43\times 10^{-3}$ (0.05/35 IMDs) was considered to be supported.").

References:

1 Kurki, M.I., Karjalainen, J., Palta, P. et al. FinnGen provides genetic insights from a well-phenotyped isolated population. *Nature* 613, 508–518 (2023).

6. The UK Biobank has been used by countless studies that focus on immune-related diseases. It is critical for this manuscript to clearly identify how this study relates to, replicates, and builds upon at least some of these studies including by identifying methodological differences.

Response:

- Thank you for your kind comments! Most of the studies concerning IMDs in UKB has focused on the genetic risk loci, ethnic distribution, identification of therapeutic targets of IMDs, comorbidities, and their shared or distinct genetic components. However, the majority of these study are based on GWAS while literature retrieval discovered that there is only a few WES of IMDs in the UKB. Notably, our study is not only a methodological innovation and extension of the previous approach to the identification of IMDs susceptibility genes/SNPs (through whole exome sequencing), but also the most

comprehensive and extensive WES of IMDs (40 IMDs in 350,770 UK Biobank participants) to date. To clarify, we have added some details in the Introduction section and discussion section.

- **Page 5, Lines 78-87:** “The UK Biobank (UKB), rich in multi-omic data, stands as an ideal platform for such endeavors, and has been widely used for sequencing studies of human diseases and traits. Prior investigations within the UKB, primarily through GWAS, have focused on genetic risk factors^{1,2}, ethnic disparities³, identification of IMDs therapeutic targets⁴⁻⁶, associated comorbidities^{7,8}, and their unique or overlapping genetic landscape⁹. However, WES studies of IMDs in the UKB have been sparse. Some have targeted specific regions, such as the HLA region for 11 autoimmune diseases¹⁰, or have investigated asthma risk mutations among predetermined variants¹¹. Additionally, many of these studies were constrained by their sample sizes; for instance, one identified the *TET2* mutation as a risk factor for gout among only 170,000 participants¹².”
- **Page 20, Line 421-426:** “The major strength of this study lies in not only the systematic investigations of protein coding variants in a series of 40 IMDs, but also the identification of a list of novel genes and exploration of their biological relevance. By virtue of focusing on coding variants, the observed associations more often provide a direct causal link between variants in a gene and a specific IMD¹³, having implications for identifying or validating drug targets.”

References:

- 1 Chia, R. *et al.* Identification of genetic risk loci and prioritization of genes and pathways for myasthenia gravis: a genome-wide association study. *Proc Natl Acad Sci U S A* **119** (2022). <https://doi.org/10.1073/pnas.2108672119>
- 2 Burren, O. S. *et al.* Genetic feature engineering enables characterisation of shared risk factors in immune-mediated diseases. *Genome Med* **12**, 106 (2020). <https://doi.org/10.1186/s13073-020-00797-4>
- 3 Sharma-Oates, A. *et al.* Early onset of immune-mediated diseases in minority ethnic groups in the UK. *BMC Med* **20**, 346 (2022). <https://doi.org/10.1186/s12916-022-02544-5>
- 4 Zhao, J. H. *et al.* Genetics of circulating inflammatory proteins identifies drivers of immune-mediated disease risk and therapeutic targets. *Nat Immunol* **24**, 1540-1551 (2023). <https://doi.org/10.1038/s41590-023-01588-w>
- 5 Lin, J., Zhou, J. & Xu, Y. Potential drug targets for multiple sclerosis identified through Mendelian randomization analysis. *Brain* **146**, 3364-3372 (2023). <https://doi.org/10.1093/brain/awad070>

- 6 Yuan, S. *et al.* Mendelian randomization and clinical trial evidence supports TYK2 inhibition as a therapeutic target for autoimmune diseases. *EBioMedicine* **89**, 104488 (2023). <https://doi.org/10.1016/j.ebiom.2023.104488>
- 7 Fromme, M. *et al.* Comorbidities in lichen planus by phenome-wide association study in two biobank population cohorts. *Br J Dermatol* **187**, 722-729 (2022). <https://doi.org/10.1111/bjd.21762>
- 8 Yuan, S. *et al.* Phenome-wide Mendelian randomization analysis reveals multiple health comorbidities of coeliac disease. *EBioMedicine* **101**, 105033 (2024). <https://doi.org/10.1016/j.ebiom.2024.105033>
- 9 Shirai, Y. *et al.* Multi-trait and cross-population genome-wide association studies across autoimmune and allergic diseases identify shared and distinct genetic component. *Ann Rheum Dis* **81**, 1301-1312 (2022). <https://doi.org/10.1136/annrheumdis-2022-222460>
- 10 Butler-Laporte, G. *et al.* HLA allele-calling using multi-ancestry whole-exome sequencing from the UK Biobank identifies 129 novel associations in 11 autoimmune diseases. *Commun Biol* **6**, 1113 (2023). <https://doi.org/10.1038/s42003-023-05496-5>
- 11 Wjst, M. Exome variants associated with asthma and allergy. *Sci Rep* **12**, 21028 (2022). <https://doi.org/10.1038/s41598-022-24960-6>
- 12 Agrawal, M. *et al.* TET2-mutant clonal hematopoiesis and risk of gout. *Blood* **140**, 1094-1103 (2022). <https://doi.org/10.1182/blood.2022015384>
- 13 Nag, A. *et al.* Effects of protein-coding variants on blood metabolite measurements and clinical biomarkers in the UK Biobank. *Am J Hum Genet* **110**, 487-498 (2023). <https://doi.org/10.1016/j.ajhg.2023.02.002>

Minor

7. It is unclear what is meant on line 73 “comprehensive and solid WES studies”. Consider rewording.

Response:

- Sorry for the inappropriate description. We have reworded this sentence to “**large-scale sequencing studies**”.

8. The term “Caucasian” is problematic and should be removed throughout. Consider following updated guidelines around ancestry and race when naming groups of people.

Response:

- Many thanks for your positive feedback towards our article. According to your suggestions and the updated guidelines around ancestry and race, we have changed the term “Caucasian” to “**European-descent**” in the revised manuscript.

9. The term “risky pattern” (e.g. line 107) should be replaced with more precise terminologies.

Response:

- Thank you for your comments! We have changed the phrase “risky pattern” to “**deleterious pattern**”.

Reviewer #3 (Remarks to the Author):

The authors present a large, systematic study of both rare and common coding variant effects on 35 immune mediated diseases (IMDs). By doing so, they discover novel disease-gene relationships and also characterise pervasive pleiotropy present amongst IMDs and other traits and biomarkers.

Overall, I think the authors results and conclusions are sound. I have several minor comments that should help increase the clarity of the presented work:

1. The paper is poorly written and has many different grammatical mistakes. For example, L56-57, L59, L107, L140, L249, just to name a few. The authors should work with the editor to ensure that this is systematically addressed before publication.

Response:

- We apologize for the language problems of our previous manuscript. The language of the entire revised manuscript has been deliberated and edited carefully by a native English speaker.

2. It was unclear to me how $P < 0.02$ (FDR) was determined? What P-value threshold does this come to, and how was it deduced?

Response:

- Thank you for your rigorous consideration. We are sorry for the inappropriate expression. In the previous manuscript, we were intended to use the exome-wide significant threshold¹⁻⁴ of $P < 2.5 \times 10^{-6}$, which is equivalent to A Benjamini-Hochberg false discovery rate (BH-FDR) threshold of < 0.02 .
- To be consistent with previous publications, we have revised the threshold to $P < 2.5 \times 10^{-6}$. Please refer to **Page 25, Lines 528-529**, “**The conventional threshold for burden analysis ($P_{SKAT} < 2.5 \times 10^{-6}$) was applied to determinant significant genes.**”

References:

- 1 Zhou W, et al. Scalable generalized linear mixed model for region-based association tests in large biobanks and cohorts. *Nat Genet.* 2020 Jun;52(6):634-639. PMID: 32424355.
- 2 Fei CJ, et al. Exome sequencing identifies genes associated with sleep-related traits. *Nat Hum Behav.* 2024 Jan 4. doi: 10.1038/s41562-023-01785-5. Epub ahead of print. PMID: 38177695.
- 3 Zhou X, et al. Integrating de novo and inherited variants in 42,607 autism cases identifies mutations in new moderate-risk genes. *Nat Genet.* 2022 Sep;54(9):1305-1319. PMID: 35982159.
- 4 Wilcox N, et al. Exome sequencing identifies breast cancer susceptibility genes and defines the contribution of coding variants to breast cancer risk. *Nat Genet.* 2023 Sep;55(9):1435-1439. PMID: 37592023.

3. Are the authors correcting for the fact they have performed 35 exome-wide association analyses? For example in the Finngen analysis, a replication P-value of 0.05 is used but the author should present replication at $P < 0.05/\text{number_of_traits_analysed}$

Response:

- Thank you for your rigorous consideration. According to your suggestions, we have applied the threshold of Bonferroni-corrected for the number of traits analyzed (0.05/35).
- We have clarified this in the Revised Manuscript (**Page 7, Lines 134-139: Results** – “Searches yielded 13 associations (19% replicated) of nominal significance (P value $< 1.43 \times 10^{-3}$; Supplementary Table 2). Notably, *FLG* for AD (OR: 2.09, $P = 6.54 \times 10^{-53}$), *ETV7* for Primary biliary cirrhosis (PBC, 1.98, $P = 2.21 \times 10^{-4}$), and *ABCG2* for gout (1.69, $P = 2.73 \times 10^{-76}$) emerged with the most pronounced coefficients...”, **Page 9, Lines 174-175: Results** – “The FinGenn dataset supported 69 of the 115 associations under the threshold of 1.43×10^{-3} (Supplementary Table 6).” and **Page 28, Lines 595-596: Methods** – “A Bonferroni-corrected threshold of $P < 1.43 \times 10^{-3}$ (0.05/35 IMDs) was considered to be supported.”).

4. Do the authors adjust for sex, age or BMI in the exome associations? IMDs are very sex specific both in terms of incidence and genetics, the authors should adjust for sex or ideally,

carry our sex-specific analyses.

Response:

- We appreciate your comments and concur with your opinions. Considering that IMDs are sex-specific diseases, all of our exome association analyses were adjusted for sex, and also age and the first 10 principal components. We have clarified in the corresponding sections in the manuscript, **Methods-Page 25, Lines 520-522**, “All association analyses incorporated covariates including sex, age, and the first 10 PCs as fixed effects”.

5. How do the authors intend to share their results with the community? Is there a website that allows researchers to query these results and download summary statistics. I insist for all reviews that both code (github) and summary level findings are made publicly available.

Response:

- Thank you for your comments. We have uploaded the codes in Github (https://github.com/Sirius-Yang/IMDs_WES). The gene-level and single variant association summary statistics of rare and common variants were uploaded in the figshare repository.

REVIEWERS' COMMENTS

Reviewer #1 (Remarks to the Author):

The manuscript has improved significantly.
The word FinnGen and the names of different diseases are not written consistently and should be correct.

Reviewer #2 (Remarks to the Author):

The authors addressed my concerns adequately.

Reviewer #3 (Remarks to the Author):

The authors have addressed all my concerns raised.

Reviewer #3 (Remarks on code availability):

The code is extremely poor, undocumented and just a collection of R, python and bash scripts without any structure. I feel the analysis would be impossible to reproduce using this code base alone.

Responses to the Reviewers

REVIEWER COMMENTS

Reviewer #1 (Remarks to the Author):

The manuscript has improved significantly.

The word FinnGen and the names of different diseases are not written consistently and should correct.

Response:

- We sincerely appreciate your feedback and are grateful to hear that the manuscript has improved significantly. In response to your comments regarding the inconsistent use of the term 'FinnGen' and the names of various diseases, we have carefully revised these throughout the document to ensure consistency. All corrected terms have been highlighted in red for easy verification. Thank you for bringing this to our attention.

Reviewer #2 (Remarks to the Author):

The authors addressed my concerns adequately.

Response:

- Many thanks for your positive feedback and we sincerely appreciate your previous suggestions which improved this article a lot.

Reviewer #3 (Remarks to the Author):

The authors have addressed all my concerns raised.

Reviewer #3 (Remarks on code availability):

The code is extremely poor, undocumented and just a collection of R, python and bash scripts without any structure. I feel the analysis would be impossible to reproduce using this code base alone.

Response:

- We are grateful for your constructive previous feedback and that on the code availability and apologize for the initial shortcomings. We have restructured our code repository to enhance reproducibility (Available at: https://github.com/Sirius-Yang/IMDs_WES/).
- Our revised submission now includes a comprehensive workflow detailing the steps and code used for each analysis and figure.
- Each key procedure is encapsulated within its own script, which is appropriately categorized into relevant directories. We have also annotated crucial sections of the code to clarify their purpose and functionality.
- Additionally, each directory now contains a README.txt file providing detailed guidance on the code and its use. To aid in visualization and understanding, we have included essential input and result files (excluding those in bed/bim/fam formats). We hope these improvements substantially facilitate the reproduction of our analyses and enhance the transparency of our research process.